# Turning a Curse into a Blessing: Enabling In-Distribution-Data-Free Backdoor Removal via Stabilized Model Inversion

**Si Chen**                                                                    *chensi@vt.edu*
*Virginia Tech*

**Yi Zeng**                                                                    *yizeng@vt.edu*
*Virginia Tech*

**Jiachen T. Wang**                                            *tianhaowang@princeton.edu*
*Princeton University*

**Won Park**                                                            *wonpark@umich.edu*
*University of Michigan*

**Xun Chen**                                                        *xun.chen@samsung.com*
*Samsung Research America*

**Lingjuan Lyu**                                                      *lingjuan.lv@sony.com*
*Sony AI*

**Zhuoqing Mao**                                                          *zmao@umich.edu*
*University of Michigan*

**Ruoxi Jia**                                                              *ruoxijia@vt.edu*
*Virginia Tech*

**Reviewed on OpenReview:** *https://openreview.net/forum?id=P880C39xAvM*

## Abstract

The effectiveness of many existing techniques for removing backdoors from machine learning models relies on access to clean in-distribution data. However, given that these models are often trained on proprietary datasets, it may not be practical to assume that in-distribution samples will always be available. On the other hand, model inversion techniques, which are typically viewed as privacy threats, can reconstruct realistic training samples from a given model, potentially eliminating the need for in-distribution data. To date, the only prior attempt to integrate backdoor removal and model inversion involves a simple combination that produced very limited results. This work represents a first step toward a more thorough understanding of how model inversion techniques could be leveraged for effective backdoor removal. Specifically, we seek to answer several key questions: What properties must reconstructed samples possess to enable successful defense? Is perceptual similarity to clean samples enough, or are additional characteristics necessary? Is it possible for reconstructed samples to contain backdoor triggers?

We demonstrate that relying solely on perceptual similarity is insufficient for effective defenses. The stability of model predictions in response to input and parameter perturbations also plays a critical role. To address this, we propose a new bi-level optimization-based framework for model inversion that promotes stability in addition to visual quality. Interestingly, we also find that reconstructed samples from a pre-trained generator's latent space do not contain backdoors, even when signals from a backdoored model are utilized for reconstruction. We provide a theoretical analysis to explain this observation. Our evalua-

tion shows that our stabilized model inversion technique achieves state-of-the-art backdoor removal performance without requiring access to clean in-distribution data, but solely by utilizing publicly available data from similar distributions. Furthermore, its performance is on par with or even better than using the same amount of clean samples.

## 1 Introduction

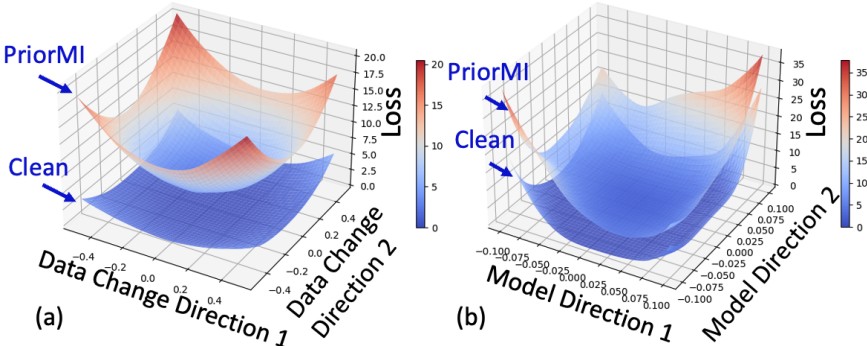

Figure 1: Loss landscape of clean data and synthetic data by a prior MI technique (Zhang et al., 2020). x, y axis are two random change directions in data space (figure (a)) and model parameter space (figure (b)); (0,0) represents the original data and model in figure (a) and (b). The synthesized data generated by prior MI is demonstrated to exhibit instability in the face of small data perturbations or changes in model parameters, unlike the stable nature observed in clean in-distribution data.

Deep neural networks have been shown to be vulnerable to backdoor attacks, in which attackers poison training data such that the trained model misclassifies any test input patched with some trigger pattern as an attacker-specified target class (Saha et al., 2020; Li et al., 2020a; Zeng et al., 2021). These attacks create a major hurdle to deploying deep networks in safety-critical applications.

Various techniques (Wang et al., 2019; Guo et al., 2019; Liu et al., 2018a) have been developed to remove the effects of backdoor attacks from a target poisoned model and turn it into a well-behaved model that does not react to the presence of a trigger. Most of the backdoor removal techniques rely on access to a set of clean samples drawn from the distribution that the poisoned model is trained on. These clean data are needed for synthesizing potential triggers and further fine-tuning the model to let the model unlearn the triggers. However, accessing clean in-distribution samples might not always be feasible. Particularly, machine learning models are often trained on proprietary datasets which are not publicly released. For instance, various ML-as-a-service platforms (e.g., Tensorflow Hub) offer trained models that users can download but often do not publish the corresponding training data.

There have been a few attempts to lift the requirement on clean in-distribution data, yet suffering unstable performance across different triggers. CLP (Zheng et al., 2022) assumes that the backdoor-related neurons in a poisoned model have a large Lipschitz constant and prunes these neurons to repair the model. However, this assumption does not hold when the trigger induces a large change in the model input. Another line of ideas (Chen et al., 2019) is to reconstruct data from the target model and then use them as a proxy for clean in-distribution data needed by existing data-reliant defenses. This line has the *unique benefit* that it can take advantage of advances of those data-reliant defenses which have already demonstrated remarkable efficacy on various triggers.

In fact, the problem of reconstructing samples from a trained model has been extensively studied in the data privacy literature, known as *model inversion (MI)*. While it is natural to utilize MI to generate data for data-reliant backdoor defense, to the best of our knowledge, there is only one work doing so: Chen et al. (2019) utilizes the simplest MI technique, which synthesizes images for a given class by optimizing the likelihood of the model for predicting that class, and further performs backdoor removal. However,

this MI technique is known to fall short in reconstructing high-dimensional input (e.g., RGB images) from a deep neural network, resulting in a noise-like pattern that lacks semantic information about a class. We observe that feeding such low-quality reconstructed samples into even the most advanced (data-reliant) backdoor removal technique leads to poor performance. Recently, a series of MI techniques have significantly improved the visual quality for high-dimensional data (Zhang et al., 2020; Chen et al., 2021; Wang et al., 2021; An et al., 2022; Struppek et al., 2022) by performing data synthesis in the latent space of a pre-trained neural network generator. Notably, this generator can be trained using publicly available data from similar distributions. These advanced techniques can often produce synthetic samples that largely retain the class-specific semantics and look perceptually similar to the original training data. However, many important questions remain unclear: *Is perceptual similarity to clean samples enough to enable successful defense, or are additional characteristics necessary? Is it possible for reconstructed samples to contain backdoor triggers?*

Intriguingly, despite the perceptual similarity between the samples synthesized by these advanced MI techniques and the original training data, we find that there is a significant gap in their resulting backdoor removal performance. Particularly, we find two factors that contribute to performance degradation (Figure 1). Firstly, we show that the model predictions at the synthesized samples are unstable to small input perturbations, which misleads downstream backdoor removal techniques to remove these perturbations instead of underlying backdoor triggers. Moreover, unlike clean samples for which the prediction loss of the model converges and thus is stable to local changes on the model parameters, the prediction loss at the synthesized samples is sensitive to small parameter changes; therefore, using them to fine-tune a poisoned model result in degradation in model accuracy. Based on these observations, we introduce an algorithmic framework for data reconstruction based on bi-level optimization, which promotes not only perceptual quality but the stability to perturbations in data and parameter space.

Moreover, the existing work Chen et al. (2019) that utilizes model inversion for backdoor removal has overlooked a critical question of whether reconstructed samples from a backdoored model could contain backdoors. Note that if the synthesized samples for the target class contain triggers, then the existing backdoor removal techniques would be nullified. Empirically, we find that as long as the pre-trained generator leveraged by MI is learned from clean data, the reconstructed samples from a poisoned model do not contain triggers. For a commonly used generator in MI literature—a generative adversarial network (GAN), we prove that backdoors are not in the range of the generator by analyzing the GAN's equilibrium.

We summarize our contributions as follows:

- We are the first to investigate the connection between MI and backdoor removal. We go beyond perceptual quality and reveal the dependence of defense performance on the stability of the inverted samples to input and parameter perturbations; and provide a theoretical understanding of why pre-trained generator-based MI does not generate backdoor-contaminated samples.

- We propose a novel bilevel optimization based data reconstruction approach for in-distribution data free defense (FRED), which maximizes the stability to input perturbation while encouraging perceptual similarity and the stability to model perturbation.

- On a range of datasets and models, employing the synthetic samples produced by FRED can lead to state-of-the-art data-free backdoor defense performance, which is comparable to or sometimes even better than using the same amount of clean data.

- FRED can be extended to match clean data's features when there is limited access to clean in-distribution samples. For instance, we show that combining just one clean in-distribution point per class with FRED can lead to a better defense performance than directly supplying 20 clean points.

## 2 Preliminaries

**Attacker Model.** Assume that an attacker performs a backdoor attack against a clean training set $D$ drawn from the distribution $\mathcal{D}$. The attacker injects a set of poisoned samples into $D$ to form a poisoned dataset $D_{\text{poi}}$. We will refer to the model trained on the poisoned dataset as a poisoned model, denoted by

$f_{\theta_{\text{poi}}}$. The goal of the attacker is to poison the training set $D$ such that for any clean test input $x$, adding a pre-defined trigger pattern $\delta$ to $x$ will change the output of the trained classifier $f_{\theta_{\text{poi}}}$ to be an attacker-desired target class $y_{\text{tar}}$. A standard technique to poison the dataset is to inject backdoored samples that are labeled as the target class and inject the trigger into their features. The model trained on such a poisoned dataset will learn the association between the trigger and the target class, thereby outputting the target class whenever a test input contains the trigger.

**Backdoor Removal.** We consider that the defender is given the poisoned model $f_{\theta_{\text{poi}}}$. The goal of the defender is to remove the effects of backdoor triggers from $f_{\theta_{\text{poi}}}$ and obtain a new model $f_{\theta*}$ that is robust to backdoor triggers, i.e., $f_{\theta*}(x + \delta) = f_{\theta*}(x)$. Many past backdoor removal techniques (including the state-of-the-art one) are based on the idea of fine-tuning the poisoned model with a set of samples, which will be referred to as the *base set*; furthermore, past techniques assume that the base set is clean and in-distribution, i.e., each sample there is drawn from $\mathcal{D}$—the distribution generating the clean portion of the data that the poisoned model is trained on. Given the base set $B = \{(x_i, y_i)\}_{i=1}^n$, Zeng et al. (2022) provides a minimax optimization framework that *unifies* a variety of different backdoor removal techniques (Wang et al., 2019; Chen et al., 2019; Guo et al., 2019):

$$\theta^* = \arg\min_\theta \max_\delta \frac{1}{|B|} \sum_{i \in B} L(f_\theta(x_i + \delta), y_i), \tag{1}$$

where the inner optimization is aimed at (approximate) *trigger synthesis*, i.e., finding a pattern that causes a high loss for predicting correct labels across all samples in the base set, and the outer optimization performs *trigger unlearning*, which seeks a model that maintains the correct label prediction $y_i$ when the synthesized trigger pattern is patched onto the input $x_i$. Zeng et al. (2022) proposed I-BAU, which achieves state-of-the-art backdoor removal performance by fine-tuning the poisoned model using mini-batch gradients of the objective in (1). Backdoor removal performance is typically measured by *attack success rate* (ASR), which measures the ratio of the backdoored samples predicted as the target class, and *clean accuracy* (ACC), which measures the ratio of the clean samples predicted as their original class. Despite the promising results, I-BAU shows that the defense performance degrades quickly as the size of available clean in-distribution samples shrinks.

**Connection between Data-Free Backdoor Removal and Model Inversion.** How to remove backdoors from a given poisoned model without access to clean, in-distribution samples? A natural idea is that as the poisoned model is trained with some clean data, it may memorize the information about the data and therefore one can potentially reconstruct the clean data from the poisoned model. Reconstructing training data from a trained model is intensively studied in the privacy literature, known as *model inversion (MI)* (Fredrikson et al., 2014; 2015). To recover training data from a given model $f_\theta$ for any class $y$, the key idea of MI is to find an input that minimizes the prediction loss of $y$:

$$x_{\text{syn}} \in \arg\min_x L(f_\theta(x), y). \tag{2}$$

DeepInspect (Chen et al., 2019) solved (2) with gradient descent for multiple times, each of which uses a randomly selected initial value of $x$; then, the base set was formed by collecting the converged input $x_{\text{syn}}$ for each initial value and pairing it with the corresponding label $y$. However, solving (2) over the high-dimensional space without any constraints generates noise-like features that lack semantic information about corresponding labels. Hence, using the samples synthesized by this way to form the base set gives unsatisfactory backdoor removal performance.

Recently, GMI (Zhang et al., 2020) proposed to optimize over the latent space of a pre-trained GAN instead:

$$x_{\text{syn}} = G(z^*), z^* \in \arg\min_z \underbrace{L(f_\theta(G(z)), y)}_{L_{\text{cl}}(z)} \underbrace{- D(G(z))}_{L_{\text{prior}}(z)}, \tag{3}$$

where $G$ and $D$ represent the generator and the discriminator of the GAN, respectively. Chen et al. (2021); An et al. (2022); Struppek et al. (2022) follow the idea of using GAN and further improve the quality of

reconstructed images with different techniques, e.g., knowledge distillation from the target model; latent space disentanglement via a StyleGAN (Karras et al., 2019; 2020), etc. These works show that the samples synthesized by the GAN-based MI technique above can maintain high visual similarity to the original training data of $f_\theta$. It is natural to ask: Can we apply these more advanced MI techniques to recover samples from the poisoned model and use them as a substitute for the clean, in-distribution samples needed in backdoor removal? Also, it is critical for the effectiveness of backdoor removal that the target-class samples in the base set do not contain backdoor triggers; otherwise, the trigger unlearning step would reinforce the association between the trigger and the target class, instead of eliminating it. Hence, another critical question is: will MI recover backdoor triggers from the poisoned model? We will answer these questions in the following section. And a more detailed discussion of related works on backdoor removal and MI can be found in Appendix A.

## 3 Using Model Inversion to Form the Base Set

MI synthesized data may suffer from visual quality degradation if the GAN is not well trained. Will better visual quality lead to a more successful defense? Besides, in Fig 1, we observe that clean data are more robust to perturbations in both data and model parameters space than GMI synthesized data; could stability also be an important property? It's important to understand key factors that contribute to removal performance and to develop a method accordingly.

To tailor backdoor removal, We formalize the optimization goal of model inversion attacks as $\min_{z_i} L_{\text{prior}}(z_i) + \lambda_1 L_{\text{cl}}(z_i) + \lambda_2 L_{\text{mp}}(z_i) + \lambda_4 L_{\text{dp}}(z_i)$. Each newly added loss term in this equation corresponds to a significant factor contributing to the backdoor removal performance. We will elaborate on the motivation for each loss term in the subsequent section.

### 3.1 Understanding Factors that Contribute to Backdoor Removal Performance

**Question 1: Does visual quality affect backdoor removal performance?** Existing MI techniques mostly focus on improving visual quality. To study how visual quality impacts backdoor removal performance, we adopt the GAN-based model inversion technique (Zhang et al., 2020) to form a base set (GMI), and follow the idea in An et al. (2022); Struppek et al. (2022) to use a StyleGAN generator to form a base set with higher quality (GMI+). We then use these base sets to perform backdoor removal with I-BAU (Zeng et al., 2022), which is the state-of-the-art backdoor removal technique. Specifically, the poisoned model is trained on a traffic sign dataset (Houben et al., 2013). The backdoor attack in Li et al. (2020a) is considered and the target class is a randomly chosen class. Figure 2 (a) shows the reconstructed samples and the original training data. We find that the samples synthesized by GMI can in general successfully recover the semantics of the clean samples, samples generated by GMI+ even achieve almost perfect visual quality. However, there is still a big gap between clean performance and GMI+ performance: the unlearning performance among different runs is quite different and the average ASR is more than 10 times higher (Figure 2 (b)). This motivates us to find other missing factors contribute to backdoor removal performance.

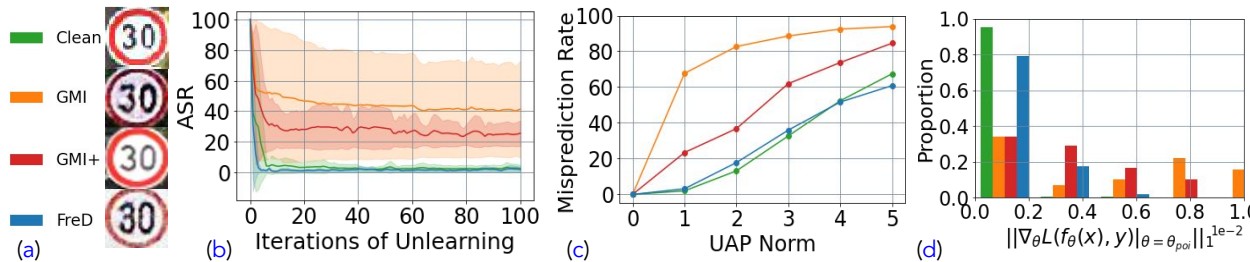

Figure 2: (a) and (b) show the example images and defense performance of the four base sets. (c) is misclassification rate when adding UAP to the four sets respectively. For each set, an optimal UAP is obtained and normalized to 1. We then gradually scale up the four UAPs and test the corresponding misprediction rate. (d) is distribution of samples given its model stability $\|\nabla_\theta L(f_\theta(x), y)|_{\theta=\theta_{\text{poi}}}\|_1$.

|  | Clean | GMI | GMI+ | FreD |
|---|---|---|---|---|
| **Trigger Norm** | 41.4484 | 32.1581 | 35.6812 | 39.2282 |

Table 1: Norm of synthesized triggers using different base sets.

**Question 2: Does stability to small perturbations affect backdoor performance?** Many backdoor removal techniques, including the state-of-the-art one, rely on trigger synthesis. Recall the backdoor unlearning optimization Eq. 1, the inner loop aims to find a trigger causing the highest loss, and the trigger is initialized to be all zeros. However, if synthetic samples are sensitive to small universal adversarial perturbation (UAP), then these UAPs could be synthesized instead of the actual trigger, leading to poor or unstable defense performance. To validate our hypothesis, we use synthesized trigger norm as a metric to evaluate stability of different base sets – a smaller synthesized trigger implies that a smaller perturbation is required to flip the prediction on a batch of samples. As shown in Table 1, clean base set leads to the largest norm. Figure 2(c) also shows that clean data need to be perturbed with a stronger UAP to reach the same misprediction rate as GMI-synthesized data, implying that clean data is more robust to UAPs.

To test whether improving stability to small pertuabtions could help decrease ASR, we introduce an additional loss in Equation 3, named data-perturbation loss: $L_{\mathrm{dp}}(z) = -\mathrm{CosSim}(f_{\theta_{\mathrm{poi}}}(G(z)), f_{\theta_{\mathrm{poi}}}(G(z) + \delta))$, where $\mathrm{CosSim}(\cdot, \cdot)$ stands for cosine similarity. This loss calculates the change of the model output logits when a synthesized sample $G(z)$ is perturbed by $\delta$. Here $\delta$ is randomly sampled from a normal Gaussian distribution. From Table 2 we can see that ASR drops when increasing weight of $L_{dp}$, which validates our hypothesis. However, $L_{dp}$ alone does not have much impact on ACC. How can we further improve the accuracy during unlearning?

|  | $L_{\mathrm{dp}}$ | | | $L_{\mathrm{mp}}$ | | |
|---|---|---|---|---|---|---|
|  | $\times\ \mathbf{10}$ | $\times\ \mathbf{100}$ | $\times\ \mathbf{1000}$ | $\times\ \mathbf{10}$ | $\times\ \mathbf{100}$ | $\times\ \mathbf{1000}$ |
| **ACC** | 0.75 | 0.82 | 0.72 | 0.94 | 0.96 | 0.97 |
| **ASR** | 0.27 | 0.08 | 0.02 | 0.42 | 0.49 | 0.37 |

Table 2: Backdoor unlearning performance with varied weights of data-perturbation ($L_{dp}$) and model-perturbation ($L_{mp}$) loss.

**Question 3: Does stability to model parameters affect backdoor performance?** The poisoned model is usually trained to be optimal on the original training data, meaning that the gradient with respect to the model parameter on the data should be close to zero: $\|\nabla_\theta L(f_\theta(x), y)|_{\theta=\theta_{poi}}\|_1 \approx 0$, as shown in Figure 1(b). However, Figure 2(d) shows that, while 90% of the clean sample has $\|\nabla_\theta L(f_\theta(x), y)|_{\theta=\theta_{poi}}\|_1 \leq 0.001$, GMI/GMI+ reconstructed samples distribute more diversely, and the gradient norm based on GMI generated samples are relatively higher. We observe that ACC quickly drops when performing unlearning on these samples, which is consistent with catastrophic forgetting phenomenon (Kirkpatrick et al., 2017). Intuitively, we can maintain a high ACC by regularizing the model gradient during inversion.

To test this hypothesis, we introduce an additional loss in Equation 3, named model-perturbation loss: $L_{\mathrm{mp}}(z) = \|\nabla_\theta L(f_\theta(G(z)), y))|_{\theta=\theta_{\mathrm{poi}}}\|_1$, which measures the stability of the prediction for the synthesized sample $G(z)$ to small changes on the parameters of the poisoned model. Table 2 shows that this loss helps improve ACC during unlearning.

## 3.2 Proposed Approach

Given the findings above, we propose FRED, an approach to reconstructing the training data from a trained model. FRED differs from recent MI techniques in that its synthesis goal not only considers synthetic data quality and recovery of class-specific semantics, but also addresses the specific challenges of non-converging prediction and small universal perturbation that hinder successful application to backdoor removal. We introduce following new loss terms critical to enable the downstream task of backdoor removal:

- The model-perturbation loss $L_{\mathrm{mp}}(z) = \|\nabla_\theta L(f_\theta(G(z)), y))|_{\theta=\theta_{\mathrm{poi}}}\|_1$.

- The data-perturbation loss $L_{\mathrm{dp}}(z, \delta) = -\mathrm{CosSim}(f_{\theta_{\mathrm{poi}}}(G(z)), f_{\theta_{\mathrm{poi}}}(G(z) + \delta))$.

- (Optional) The feature consistency loss $L_{\mathrm{con}}(z) = \sum_{(x', y') \in D_{\mathrm{clean}}, y' = y} \|g_{\theta_{\mathrm{poi}}}(G(z)) - g_{\theta_{\mathrm{poi}}}(x')\|_2$. $g_{\theta_{\mathrm{poi}}}$ represents the feature extractor of the poisoned model $f_{\theta_{\mathrm{poi}}}$, i.e., the output of the penultimate layer. The loss is only used when we extend our approach to defense setting where a set of clean in-distribution samples $D_{\mathrm{clean}}$ is available. It measures the feature distance between the synthesized sample and the available clean samples.

---

**ALGORITHM 1:** Algorithm of FRED.

**Input** : Generator $G$, target model $T$, batch size $B$, clean data $x$ (optional), max iterations $N$, learning rate $\alpha_1, \alpha_2$.

**1 for** *each class* $y \in (1, K)$ **do**

**2**     Initialize $z$: $z^{(1)} \sim \mathbb{N}(0, I)$.

**3**     Initialize $\delta$: $\delta^{(1)} = \mathbf{0}^{1 \times d}$ where $d$ is the dimension of synthesized sample $G(z)$.

**4**     **for** *each iteration* $i \in (1, N)$ **do**

**5**        Temporary Update $z$: $\hat{z}^{(i)} = z^{(i)} - \alpha_1 \frac{1}{B} \sum_{b=1}^{B} \nabla_z L_{dp}(z^{(i)}, \delta^{(i)} | y, x)$.

**6**        Update $\delta$: $\delta^{(i+1)} = \delta^{(i)} + \alpha_2 \frac{1}{B} \sum_{b=1}^{B} \nabla_\delta L_{dp}(\hat{z}^i, \delta^{(i)})$.

**7**        Update $z$: $z^{(i+1)} = z^{(i)} - \alpha_1 \frac{1}{B} \sum_{b=1}^{B} \nabla_z L_{total}(z^{(i)}, \delta^{(i+1)} | y, x)$.

**8**     **end**

**9**     $z_y = z^{(N)}$

**10 end**

**11 return** $z_1, \ldots, z_K$

---

Instead of using simple Gaussian noise in $L_{dp}$, we propose a bilevel-optimization algorithm 1 to find the most potent universal perturbation for $B$ synthesized samples:

$$
\begin{aligned}
\delta^* &= \arg\max_\delta \sum_{i=1}^{B} L_{\mathrm{dp}}(z_i^*(\delta), \delta) \\
\text{s.t.} \quad z_i^*(\delta) &= \arg\min_{z_i}\{L_{\mathrm{prior}}(z_i) + \lambda_1 L_{\mathrm{cl}}(z_i) + \lambda_2 L_{\mathrm{mp}}(z_i) + \lambda_3 L_{\mathrm{con}}(z_i) + \lambda_4 L_{\mathrm{dp}}(z_i, \delta)\} \\
&\forall i \in \{1, \ldots, B\}
\end{aligned}
\tag{4}
$$

However, it could be computationally expensive as the inner optimization at any $\delta$ requires synthesizing a batch of samples. To tackle this challenge, we propose an online approximation algorithm 1 to update $z$ and $\delta$ alternatively through a single optimization loop. This type of algorithm is often used in meta learning (Shu et al., 2019; Madaan et al., 2021). The output of 1 provides the optimal perturbation and a batch of synthesized samples robustified against the perturbation.

**Hyperparameter Tuning:** $\lambda_1$ to $\lambda_4$ are the weights associated with each loss in Equation 4. In our experiments, we choose $\lambda_1 = 1000$ following the prior works on MI. We find $\lambda_3 = 1000, \lambda_4 = 10$ work well across different datasets and models. The best value for $\lambda_2$ is task-dependant, chosen by grid search that yields the smallest value of $L_{\mathrm{prior}} + L_{\mathrm{cl}} + L_{\mathrm{mp}} + L_{\mathrm{dp}}$. Table 2 provides a sensitive analysis of $\lambda_2$ and $\lambda_4$, which are the weights of our proposed loss terms. Detailed choice of other hyperparameters are provided in Appendix C.

## 4 MI does not recover backdoors

### 4.1 Empirical Study

We perform experiments on the GTSRB dataset (Houben et al., 2013) to study whether the synthesized samples would contain backdoors. Specifically, we train poisoned models under various attacks e.g., invisible attack (Li et al., 2020a), smooth (Zeng et al., 2021), trojan square (Liu et al., 2018b), WaNet (Nguyen & Tran, 2020b); and apply FRED to reconstruct a set of samples from this model. We generate 100 images

for each class. To detect whether or not our synthesized data contain the backdoor trigger, we train a binary trigger detection classifier on a clean GTSRB training set, a GAN-generated set, and their poisoned correspondence. The trained trigger classifier has 100% accuracy on a held-out test set. Applying this trigger classifier to our synthesized data, we get that no images are detected to contain the trigger. However, it is still possible that backdoored data are within the support of the GAN generated images but just not discovered by our synthesis technique as the underlying optimization does not directly lead the synthesized data to recover the backdoored samples. To empirically verify whether the backdoored data are not found by our optimization or they are not in the support of the GAN, after regular optimization as shown in Algorithm 1, we continue to optimize these synthesized data to minimize the mean square error (MSE) between them and their poisoned version. However, even after the optimization, backdoor detection rate is still 0. The intuition here is to encourage GAN to generate trigger pattern by maximizing the pixel-level similarity between GAN generated images and poison images. We adopted MSE as our loss function because it's commonly used for pixel-level recovery in the literature. While cosine similarity can also be used to measure the pixel-level similarity, it is not easy to optimize as MSE.

Above experiments are all based on the assumption that the auxiliary dataset $D_{\mathrm{aux}}$ used for training the GAN is clean. What if some poisoned data is mixed into the GAN's training data? We further poison $D_{\mathrm{aux}}$ with different ratio and test the trigger detection rate respectively. As shown in Figure 3, detection rate remains 0 when $D_{\mathrm{aux}}$ is poisoned by invisible attack (Li et al., 2020a) using 1% and 2% poison rate, which is typically used for backdoor attacks. Even when the poison rate is increased to an uncommonly large ratio (i.e., 50%), the detection rate remains low at 0.02. Bau et al. (2019) shows similar findings of the wholesale omission of GAN on generating some objects. These objects are either complex patterns that include many pixels (e.g., large human figure), or are under-represented in the GAN training distribution. Based on these findings, even if backdoors exist in GAN's training data, it would be hard for them to take effect unless using a substantial poison rate, which becomes unrealistic. Also, recent backdoor data detection methods (Ma et al., 2022) have achieved great efficacy. They can serve as pre-processing steps to screen out poisoned samples in GAN's training set.

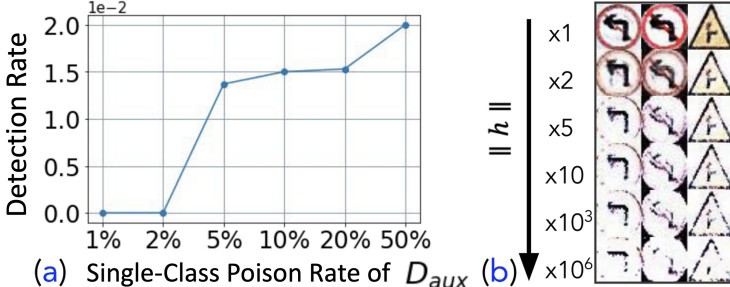

Figure 3: (a) Trigger detection rate when increasing data poison rate of samples from a single class in auxiliary dataset $D_{aux}$. Trigger detection rate here refers to the percentage of images that were detected containing triggers among all generated images, while the poison rate refers to the percentage of poison images with the target-class images only. (b) Example of images generated from latent code $h$ with different scales of norm.

## 4.2 Theoretical Justification based on Equilibrium in GAN

Here, we provide the theoretical justification of why reconstructing samples from a poisoned model based on a (clean) pre-trained GAN does not recover backdoored samples. We require the following two assumptions.

**Assumption 1.** *The generator $G$ is $L$-Lipschitz in latent vector input $h \in \mathbb{R}^p$.*

**Assumption 2.** *For all $h \in \mathbb{R}^p$, if $\|h\| \geq B$ for some $B > 0$, then $G(h)$ has no semantic meaning.*

The Assumption 2 is justified by Figure 3 (b): We observe an apparent quality degradation of the generated images when increasing the norm of the latent vector $h$. When the norm is $10^6$ larger, the generated images

do not contain semantic meaning. Note that both clean and backdoor data are considered to have semantic meaning.

We are interested in whether the range of the generator range($G$) contains backdoored data. Since the backdoored images still have semantic meaning by definition, if range($G$) contains backdoored data they will be within the high-density region where the corresponding latent vector $h$ has $\|h\| \leq B$. By the Lipschitzness (which implies continuity) of $G$, it means the density of the distribution induced by the generator ($G(h), h \sim \mathcal{N}(0, I)$) on the backdoored data points $> 0$.[1] Thus, we reduce the question of "*whether the range of $G$ contains backdoored data*" to "*whether the generator distribution has a non-zero density on backdoored data points*."

We proceed by formulating GAN training as a two-player game between the generator and discriminator. The game terminates only when the two players reach a min-max solution where neither party has the incentive to deviate from the current state. Such a min-max solution is called *pure Nash equilibrium*. Based on the game-theoretic framework, we show the following result, and the proof is outlined in Appendix B.

**Theorem 3** (Informal). *When the generator learns a distribution with non-negligible density on backdoored data, the generator and discriminator cannot achieve pure strategy Nash equilibrium.*

This result implies that, no backdoored data point can appear in range($G$) when the GAN is trained properly where the generator and discriminator reach equilibrium. Thus, no matter how we search over $G(h)$ for different latent vector $h$ during the MI step, it is impossible to find an $h$ such that $G(h)$ is a backdoored image.

## 5 Evaluation

Our evaluation focuses on the following aspects:

- Assess the effectiveness of FRED to enable data-free backdoor removal, and the benefit of FRED with a limited amount of clean samples available (Sec. 5.2, Sec. D);
- Study application of FRED to adversarial fine-tuning (Jeddi et al., 2020), which aims to robustify a pre-trained model against adversarial examples (Goodfellow et al., 2014) (Sec. 5.2, Sec. D);
- Ablation study on several design choices of FRED, including different loss terms and the number of synthesized samples (Sec. 5.2);
- Study performance of FRED with varing distribution shifts OOD datasets (Sec. C.1);
- Study the effectiveness of FRED supplying other existing backdoor cleansing methods (Sec. E).
- Study the effectiveness of FRED against adaptive attacks (Sec. F).

Our evaluation setting reflects real-world scenarios, where the training data of pre-trained GANs often differs from the target model's training set in terms of distribution (e.g., pre-trained GAN is usually built with large, open data, yet the target model could be built by some stakeholder with proprietary datasets). We also considered out-of-distribution data with various distributional shifts from the target-model training data to enable a comprehensive evaluation (refer to Section C.1).

### 5.1 Experimental Setup

**Data.** We evaluate datasets built for different prediction tasks, including face recognition, traffic sign classification, and general object recognition. For each task, we choose two datasets, one used for training the poisoned model and another for learning a pre-trained GAN. Detailed usage of the datasets is shown in Table 3.

**Backdoor Attacks.** We evaluate nine different kinds of backdoor attacks in all-to-one settings (the target model will misclassify all other classes' samples patched with the trigger as the target class), including the

---

[1]Note that the set of latent vectors with semantic meaning $\{h : \|h\| < B\}$ is an open set.

| | Face Recognition | Traffic Sign Classification | General Object Detection |
|---|---|---|---|
| **Poisoned Model** | PubFig(Pinto et al., 2011) | GTSRB (Houben et al., 2013) | CIFAR-10 (Krizhevsky et al.) |
| **Pre-trained GAN** | CelebA (Liu et al., 2015) | TSRD(Huang) | STL-10 (Coates et al., 2011) |

Table 3: Datasets.

hidden trigger backdoor attack (Hidden) (Saha et al., 2020), input-aware backdoor (IAB) attack (Nguyen & Tran, 2020a), WaNet (Nguyen & Tran, 2020b), $L_0$ invisible ($L_0$ inv) (Li et al., 2020a), $L_2$ invisible ($L_2$ inv) (Li et al., 2020a), the frequency invisible smooth (Smooth) attack (Zeng et al., 2021), trojan watermark (Troj-WM) (Liu et al., 2018b), trojan square (Troj-SQ) (Liu et al., 2018b), and blend attack (Chen et al., 2017). The implementation details of the attacks are deferred to Appendix C.

**Baselines.** We compare FRED with five baselines, where the first four baselines differ in kinds of samples contained in the base set and share the same downstream backdoor removal technique, namely I-BAU, which achieves the state-of-the-art backdoor removal performance given a clean base set. 1) Clean: The base set is formed by clean samples drawn from the original training data of the poisoned model. 2) Out-of-the-distribution (OOD): The base set consists of **ALL** of the OOD samples that are used for learning the pre-trained GAN. 3) Naive: The base set contains samples synthesized by the MI adopted in Chen et al. (2019) which directly optimizes in the pixel space. 4) GMI: The base set is formed by the synthetic samples from GMI (Zhang et al., 2020). The comparison between FRED and GMI will demonstrate the effectiveness of our designed loss terms. 5) CLP (Zheng et al., 2022): The last baseline is a recent data-free backdoor removal technique that does not utilize the idea of data synthesis. Instead, it prunes the neurons directly based on corresponding Lipschitz constants.

Our baselines provide comparison from different perspectives: 1) Clean is the original backdoor removal method with an extra clean in-distribution set provided, serving as the upper bound for defense performance. 2) OOD, Naive, GMI share the same assumption as our defense that the defender can leverage clean publicly available data from a related domain. This setting is practical in real-world scenarios considering the large amount of public data available online. 3) CLP is the state-of-the-art backdoor removal defense that is not base-set-reliant compared to this type of baselines, our method offers a unique plug-and-play benefit.

**Protocol.** For all baselines except OOD, we draw/ generate 20 samples per class for PubFig and GTSRB; 40 for CIFAR-10. A detailed study of choosing the number of samples to be generated for each class is shown in Section 5.2. For the hyperparameters, we fix $\lambda_1 = 1000, \lambda_3 = 1000, \lambda_4 = 10$, set $\lambda_2 = 10$ for PubFig and GTSRB, and $\lambda_2 = 100$ for CIFAR-10. The defense performance is averaged over three random-initialized runs of I-BAU.

| | | | $L_0$ inv | | | | | | | $L_2$ inv | | | | |
|---|---|---|---|---|---|---|---|---|---|---|---|---|---|
| | Initial | Clean | OOD | Naive | GMI | FreD | CLP | Initial | Clean | OOD | Naive | GMI | FreD | CLP |
| **ACC** | 0.97 | 0.98 | 0.88 | 0.88 | 0.93 | **0.95** | 0.94 | 0.97 | 0.98 | 0.9330 | 0.95 | 0.94 | **0.94** | 0.94 |
| **ASR** | 1.0 | 0.03 | 0.02 | 0.08 | 0.09 | **0.02** | 0.02 | 0.998 | 0.06 | 0.832 | 0.06 | 0.48 | **0.01** | 0.01 |
| | | | **Smooth** | | | | | | | **Wanet** | | | | |
| | Initial | Clean | OOD | Naive | GMI | FreD | CLP | Initial | Clean | OOD | Naive | GMI | FreD | CLP |
| **ACC** | 0.97 | 0.98 | 0.83 | 0.82 | 0.96 | **0.97** | 0.18 | 0.98 | 0.94 | 0.26 | 0.17 | 0.81 | **0.94** | 0.01 |
| **ASR** | 0.998 | 0.1 | 0.40 | 0.05 | 0.03 | **0.02** | 0.02 | 0.99 | 0.05 | 0.34 | 0.97 | 0.16 | **0.05** | 0.99 |
| | | | **IAB** | | | | | | | **Troj-SQ** | | | | |
| | Initial | Clean | OOD | Naive | GMI | FreD | CLP | Initial | Clean | OOD | Naive | GMI | FreD | CLP |
| **ACC** | 0.94 | 0.97 | 0.81 | 0.45 | 0.89 | **0.91** | 0.92 | 0.98 | 0.96 | 0.40 | 0.45 | 0.86 | **0.94** | 0.81 |
| **ASR** | 1.0 | 0.02 | 0.10 | 0.11 | 0.11 | **0.10** | 0.10 | 1.0 | 0.01 | 0.06 | 0.16 | 0.10 | **0.06** | 0.23 |
| | | | **Troj-WM** | | | | | | | **Blend** | | | | |
| | Initial | Clean | OOD | Naive | GMI | FreD | CLP | Initial | Clean | OOD | Naive | GMI | FreD | CLP |
| **ACC** | 0.98 | 0.96 | 0.47 | 0.62 | 0.84 | **0.87** | 0.87 | 0.98 | 0.96 | 0.47 | 0.68 | 0.82 | **0.92** | 0.75 |
| **ASR** | 1.0 | 0.01 | 0.30 | 0.09 | 0.30 | **0.09** | 0.12 | 1.0 | 0.08 | 0.51 | 0.55 | 0.48 | **0.22** | 0.85 |

Table 4: Results of FRED boosted backdoor unlearning on GTSRB.

### 5.2 Results

**Data-Free Backdoor Defense.** Table 4 shows that FRED outperforms naive MI, OOD, GMI and CLP against various backdoor attacks on GTSRB. Results for the other datasets can be found in Appendix D and FRED remains the best. Figure 7 visualizes the samples synthesized by Naive, GMI, and FRED. GMI, and FRED can generate samples with better visual quality, whereas Naive generates merely noise-like samples. This visualization explains the significant defense performance improvement achieved by GMI and FRED upon Naive. The performance of FRED is mostly on par with Clean. Interestingly, FRED achieves a higher ACC and comparable ASR than baseline utilizing clean data when defending against the IAB attack performed on the CIFAR-10 dataset. This may be because the model is overfitted to the clean training samples, and samples generated by FRED reduce the degree of overfitting by providing more abundant features. Note that CLP fails in defending against Smooth, Wanet, and Blend attack: Either the ACC drops to close to zero (Smooth and Wanet), or ASR remains high (Wanet and Blend), or both occurs (Wanet). As the above triggers have large norm and hence induce large changes in the model input, CLP's assumption that backdoor-related neurons in the poisoned model have a large Lipschitz constant does not hold. To better interpret the data synthesis process of FRED, we show a series of samples generated at different iterations in Figure 6. We observe that the appearance of the generated images varies significantly over the first ten optimization iterations and stabilizes afterwards.

**Data-Limited Backdoor Defense.** Here, we evaluate the benefits of FRED when there exists a tiny amount of clean samples. Particularly, we consider a stress test with 1 sample. With a single sample, even the state-of-the-art data-reliant backdoor removal technique works poorly as shown in the CIFAR-10 and PubFig83 results in Table 5). To evaluate FRED, we use FRED with the proposed feature consistency loss $L_{\text{con}}$ to generate 20 additional samples for each class, and the final result (FRED-Booster) is obtained by using the combination of both 1 clean sample and 20 generated samples for each class. Table 5 shows that FRED can significantly boost the defense performance compared to solely using the available clean sample(s). Moreover, using 20 samples from FRED plus one clean sample gives better defense performance than 20 clean samples. As a final note, compared to FRED, CLP, the data-free backdoor removal baseline based on model pruning, cannot be benefited from additional clean samples.

|  |  | Clean(20) | Clean(1) | FreD-Booster |
|---|---|---|---|---|
| **GTSRB** | **ACC** | 0.98 | 0.93 | 0.98 |
| **Smooth** | **ASR** | 0.01 | 1 | 0 |
| **CIFAR-10** | **ACC** | 0.82 | 0.52 | 0.85 |
| **IAB** | **ASR** | 0.03 | 0.18 | 0.01 |
| **PubFig** | **ACC** | 0.86 | 0.44 | 0.86 |
| **Troj-WM** | **ASR** | 0.03 | 0.35 | 0 |

Table 5: Results of backdoor unlearning performance with a small amount of clean data and generated samples.

|  |  | Initial | GMI | $L_{mp}$ | $L_{dp}$ | $L_{mp} + L_{dp}$ |
|---|---|---|---|---|---|---|
| **GTSRB** | **ACC(%)** | 98.8 | 86.52 | 92.70 | 89.32 | 94.49 |
|  | **ASR(%)** | 100.0 | 10.06 | 8.63 | 8.25 | 5.90 |
| **PubFig** | **ACC(%)** | 92.21 | 70.52 | 72.32 | 70.61 | 83.53 |
|  | **ASR(%)** | 100.0 | 26.06 | 3.25 | 1.63 | 2.88 |

Table 6: Ablation study of proposed model-perturbation loss $L_{mp}$ and data-perturbation loss $L_{dp}$.

**Ablation of Loss Terms.** We proposed two loss terms 1) model-perturbation loss $L_{\text{mp}}$ and 2) data-perturbation loss $L_{\text{dp}}$ to improve the utility of the synthesized samples in the data-free backdoor defense setting. Table 6 presents an ablation study of the two losses on a poisoned model trained on GTSRB under the trojan square attack as well as a model trained on Celeba under the trojan watermark attack. We observe that $l_{\text{dp}}$ improves the ASR more than $l_{\text{mp}}$ while $l_{\text{mp}}$ is a more critical driver of maintaining the ACC

compared to $l_{dp}$. This observation aligns with our design objectives. Recall that $l_{dp}$ is designed to enable effective synthesis of backdoor trigger and thus directly related to the reduction of ASR. On the other hand, $l_{mp}$ encourages the stability of prediction to small parameter changes, which in turn mitigates catestrophic forgetting during unlearning; hence, it is directly related to maintaining the clean accuracy.

**Ablation of Base Set Sizes.** We study the impact of the number of the synthesized samples contained in the base set. We choose the number of samples for each class to be $[1, 5, 10, 15, 20, 25, 30]$ and evaluate the defense performance by averaging over 3 runs of the defense. To better interpret the performance of FRED, we also compare with Clean, Naive and GMI using the same amount of samples. Note that this experiment excludes the OOD baseline: we use the poisoned model to generate pseudo-labels for the OOD samples but because the label space of the OOD samples and that of the poisoned model may not overlap, the labeled OOD samples are insufficient or even none for some classes. Figure 4 shows that the defense performance keeps increasing as the number of samples increases and converges to optimum when the number of samples for each class is above 20. FRED, GMI, and Clean can maintain a high ACC when a larger number of generated samples are used, but Naive suffers a significant ACC drop. We also observe during the experiments that performance of Naive has a large variance when evaluated over base set with different size. The variance could induce from the inconsistent quality among generated samples, or instability of the samples against input perturbation, leading to synthesizing inaccurate triggers. On the other hand, the variance of FRED is similar to the variance of Clean, indicating good generalizability of FRED-enabled defenses. Another interesting finding is that when performing unlearning with a small amount of samples (i.e., 1 or 5 per class), FRED even achieves higher ACC than Clean.

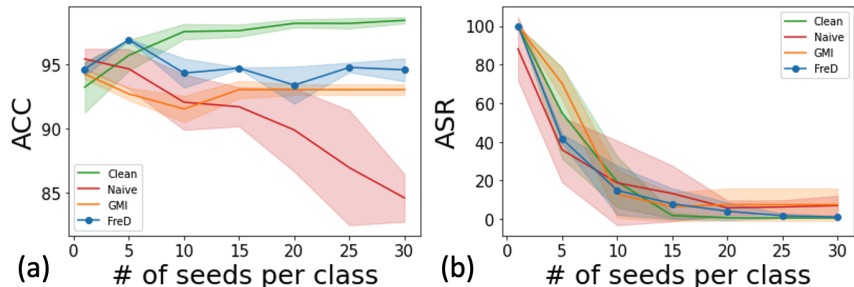

Figure 4: Ablation study of the number of samples used on backdoor defenses on the GTSRB with $L_0$ inv attack.

## 6 Conclusion

In this paper, we investigate the connection between model inversion and backdoor defense, and present FRED to generate synthetic samples that can be used as a substitute for clean in-distribution data to support backdoor removal. FRED can also be used to boost defense performance when only limited clean in-distribution data are available. This work sets a foundation towards developing highly effective in-distribution-data-free backdoor defenses. In particular, one can potentially supply our synthetic data to other future defenses to enable their data-free mode of usage or improve their performance in the limited data setting.

## 7 Acknowledgement

RJ and the ReDS lab acknowledge support through grants from the Amazon-Virginia Tech Initiative for Efficient and Robust Machine Learning, the National Science Foundation under Grant No. IIS-2312794, NSF IIS-2313130, NSF OAC-2239622, and the Commonwealth Cyber Initiative. JTW is supported by Princeton's Gordon Y. S. Wu Fellowship.

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

# A   Related Works

**Backdoor Defenses.**   Backdoor defenses normally can be performed on two levels: data-level and model-level. For data-level detection or cleaning, the defender aims to identify (Gao et al., 2019; Chen et al., 2018; Tran et al., 2018; Koh & Liang, 2017; Chou et al., 2020; Zeng et al., 2021) or purify (Doan et al., 2020) the poison input in the training set, thus requiring the access to training data. Model-level detection (Liu et al., 2019; Shen et al., 2021) or cleaning, instead, aims to detect if a pre-trained model is poisoned or mitigate vulnerabilities of the models. In this paper, we focus on model-level cleaning. Most of the prior works on this line (Wang et al., 2019; Chen et al., 2019; Guo et al., 2019; Zeng et al., 2022; Li et al., 2020b; Borgnia et al., 2020; Qiu et al., 2021) require a small set of clean data to synthesize triggers and further perform unlearning. Among them, I-BAU (Zeng et al., 2022) has achieved the state-of-the-art defense performance against a wide range of existing attacks. However, the performance of I-BAU degrades as the number of clean samples reduces. We aim to enable I-BAU to function effectively without any clean data. A recent work (Zheng et al., 2022) proposes a method to perform backdoor removal without using clean data. They identify model channels with high Lipschitz constants, which are directly calculated from the weight matrices, as backdoor related channels; and do simple pruning to repair the model. However, their method only applies to backdoor scenarios and cannot benefit from clean samples if available. Our method, by contrast, also applies to evasion attacks and is able to leverage available clean samples to further boost defense performance. Above all, the performance of our method is more favorable.

**Model Inversion.**   The goal of model inversion (MI) is similar to ours. But from an attack perspective, MI aims to divulge sensitive attributes in the training data, and to achieve this goal, the generated data should have good visual quality. Fredrikson *et al.* (Fredrikson et al., 2015) follows the maximum likelihood principle and performs MI by searching over the image space for a sample with highest likelihood under the given target model. DeepInspect (Chen et al., 2019) employs this simple MI to generate a surrogate training set for backdoor unlearning and achieves good results on MNIST and GTSRB. However, we find that samples generated by this naive MI approach have bad visual quality and usually fail in downstream defenses on high-dimensional datasets (e.g., PubFig and CIFAR-10). In this paper, we build upon the idea of recent MI works (Zhang et al., 2020; Chen et al., 2021) that search for a synthetic sample in the latent space of a pre-trained GAN instead of the image space. Even when the GAN is not trained on the in-distribution data, this idea can greatly help improve the visual quality of synthesized samples. The key innovations that set our work apart from the MI attacks is that we go beyond the traditional "high-likelihood" assumption made in all existing MI works about clean data and further formalize other plausible assumptions, especially those related to data- and model-stability. We show that enforcing the synthetic data to satisfy these assumptions can significantly improve their utility for defenses.

# B   Why is backdoored data not on GAN's range?

**Notations.**   Throughout the section, we use $d$ for the dimension of samples, and $p$ for the dimension of the latent vector. We denote discriminator $D : \mathbb{R}^d \to [0, 1]$, generator $G : \mathbb{R}^p \to \mathbb{R}^d$. We use $\mathcal{P}_{\text{real}}$ to denote the real distribution the GAN aims to learn. The generator $G$ defines a distribution $\mathcal{P}_G$ as follows: generate latent vector $h \sim \mathcal{N}(0, I_p)$ from $p$-dimensional spherical standard Gaussian distribution, and then apply $G$ on $h$ and generate a sample $x = G(h)$. We denote the class of discriminators as $\mathcal{D} = \{D\}$ and the class of generators $\mathcal{G} = \{G\}$. Ideally, $\mathcal{D}$ is the class of all 1-Lipschitz functions. For a distribution $\mathcal{P}$, we use $\mathcal{P}(x)$ to denote the density of $\mathcal{P}$ on $x$, and $\mathcal{P}(S)$ denotes the Lebesgue integration $\int I[x \in S] d\mathcal{P}(x)$. We use $\mathbb{E}_{\mathcal{P}}[D]$ as an abbreviation for $\mathbb{E}_{x \sim \mathcal{P}}[D(x)]$. We use supp($\cdot$) to denote the support of distribution. We use $d_{\text{W}}(\cdot, \cdot)$ to denote 1-Wasserstein distance with $\ell_2$-norm, i.e., the Earth Mover distance.

The two required assumptions are demonstrated in previous section 4.2. To formally state our theorem, we formulate the training of GAN as a game between generator and discriminator.

**Definition 4** (Payoff)**.** *For a class of generators $\mathcal{G} = \{G\}$ and a class of discriminators $\mathcal{D} = \{D\}$, we define the* payoff *$F(D, G)$ of the game between generator and discriminator as*

$$F(D, G) = \mathbb{E}_{x \sim \mathcal{P}_{\text{real}}} [D(x)] - \mathbb{E}_{x \sim \mathcal{P}_G} [D(x)] \tag{5}$$

The generator and discriminator aims at reaching a min-max solution, i.e., the *pure Nash equilibrium*, where neither party has the incentive to deviate from the current state..

**Definition 5** (pure equilibrium). *A pair of strategy $(D^*, G^*)$ is a pure equilibrium if for some value $V$,*

$$\forall D \in \mathcal{D}, F(D, G^*) \leq V$$
$$\forall G \in \mathcal{G}, F(D^*, G) \geq V$$

However, such an equilibrium may not be achievable for a pure strategy setting. We introduce a natural relaxation for quantifying the extent of equilibrium between a pair of generator/discriminator.

**Definition 6** ($\varepsilon$-approximate pure equilibrium). *A pair of strategy $(D^*, G^*)$ is an $\varepsilon$-approximate pure equilibrium if for some value $V$,*

$$\forall D \in \mathcal{D}, F(D, G^*) \leq V + \varepsilon$$
$$\forall G \in \mathcal{G}, F(D^*, G) \geq V - \varepsilon$$

We are now ready to state our main results.

**General Idea: game-theoretic interpretation of GAN.** From a game theory perspective, the generator and discriminator in GAN are engaged in a two-player zero-sum game. In game theory, a Nash Equilibrium is a state in which neither player can improve their own outcome by changing their strategy, assuming the other player keeps their strategy fixed. In the context of GANs, a Nash Equilibrium is achieved when the Generator produces data that is indistinguishable from real data, and the Discriminator is no better than random guessing at telling the difference between real and generated data. In Theorem 3 (or Theorem 7), we show that when the generator's learned distribution contains a significant probability mass on backdoored images, which is out-of-distribution (OOD) data, the equilibrium cannot be established between the generator and discriminator, which forces the generator to reduce the probability mass on the backdoored images.

**Theorem 7** (Formal version of Theorem 3). *Given any two distributions $\mathcal{P}_1, \mathcal{P}_2$ s.t. for the set $S_{\text{OOD}} = \{x \in supp(\mathcal{P}_2) : \min_{y \in supp(\mathcal{P}_1) \cup supp(\mathcal{P}_{\text{real}})} \|x - y\| \geq 1\}$, we have $\mathcal{P}_2(S_{\text{OOD}}) \geq 1 - q'$ for some $q' \in [0, 1)$. Let $\mathcal{D}^* = \arg\max_{D \in \mathcal{D}} \mathbb{E}_{\mathcal{P}_{\text{real}}}[D] - \mathbb{E}_{\mathcal{P}_1}[D]$ and $D^* \in \mathcal{D}^*$. If $G$ induce a mixture distribution $\mathcal{P}_G = (1-q)\mathcal{P}_1 + q\mathcal{P}_2$ for some $q \in (0, 1)$, then there exists no $D \in \mathcal{D}$ s.t. $(D, G)$ is $\varepsilon$-approximate pure equilibrium for any $\varepsilon < \frac{1}{2}q(\mathbb{E}_{\mathcal{P}_1}[D^*] - q')$. Moreover, when $\mathcal{P}_{\text{real}} = \mathcal{P}_1$, we have $\varepsilon < \frac{1}{2}q(1 - q')$. Further more, given Assumption 1 and 2, we can lower bound $q$ if $\text{range}(G)$ contains backdoored data, which leads to $\varepsilon < \frac{1}{2}(1 - q') \left(\frac{1}{L\sqrt{2}}\right)^p \frac{\exp(-\frac{1}{2}B^2)}{\Gamma(p/2+1)}$, where $\Gamma$ is the Gamma function.*

**Interpretation.** To interpret the above theorem statement, one can regard $\mathcal{P}_1$ as a clean distribution (not necessarily $\mathcal{P}_{\text{real}}$), and $\mathcal{P}_2$ as a distribution that contains backdoor data on its support. Since backdoored images are separated from clean images (i.e., out-of-distribution (OOD) data), we can assume that all backdoored images are within the set $S_{\text{OOD}} = \{x \in \text{supp}(\mathcal{P}_2) : \min_{y \in \text{supp}(\mathcal{P}_1) \cup \text{supp}(\mathcal{P}_{\text{real}})} \|x - y\| \geq 1\}$. The above theorem thus states that no equilibrium could be achieved if $\mathcal{P}_G$ has non-negligible density on $S_{\text{OOD}}$. The theorem essentially reveals the tension between a generator's goal to trick the discriminator and its propensity to generate backdoored (OOD) data. In simple terms, think of $\mathcal{P}_1$ as a distribution representing normal, clean images, and $\mathcal{P}_2$ representing malicious, backdoored images. The generator mixes these distributions to create $\mathcal{P}_G$, which it uses to try to fool the discriminator. Now, if the $\mathcal{P}_G$ mixture has a significant probability of sampling backdoored (OOD) images from $\mathcal{P}_2$, it is much easier for the discriminator to spot the differences between the generated and real data. Hence, there exists a better move for the generator to increase its "payoff" by sampling less from $\mathcal{P}_2$, hence an equilibrium becomes unattainable under such $\mathcal{P}_G$ (captured by the upper bound of $\varepsilon$). In essence, the generator is in a quandary: including backdoored data in the mixture makes it easier for the discriminator to win, which is contrary to what the generator wants.

**Remark.** *It is also possible that $\mathcal{P}_1$ also supports on $\{x : \min_{y \in supp(\mathcal{P}_{\text{real}})} \|x - y\| \geq 1\}$, but this leads to vacuous results.*

## B.1 Proof of The Formal Theorem

**Proof Overview.** In Lemma 8, we derive the upper bound of the degree of equilibrium $\varepsilon$ when $\mathcal{P}_G = (1-q)\mathcal{P}_1 + q\mathcal{P}_2$ in terms of the best discriminator in distinguishing between $\mathcal{P}_{\text{real}}$ and $\mathcal{P}_1$, i.e., $\mathcal{D}^* = \arg\max_{D \in \mathcal{D}} \mathbb{E}_{\mathcal{P}_{\text{real}}}[D] - \mathbb{E}_{\mathcal{P}_1}[D]$. Then in Lemma 9 and 10, we further bound different terms in the upper bound of $\varepsilon$ derived in Lemma 8 under Assumption 1 and 2 stated in the maintext.

**Lemma 8.** *Given any two distributions $\mathcal{P}_1, \mathcal{P}_2$, let $\mathcal{D}^* = \arg\max_{D \in \mathcal{D}} \mathbb{E}_{\mathcal{P}_{\text{real}}}[D] - \mathbb{E}_{\mathcal{P}_1}[D]$. For any $D^* \in \mathcal{D}^*$, if $G$ induce a distribution $\mathcal{P}_G = (1-q)\mathcal{P}_1 + q\mathcal{P}_2$, then there exists no $D \in \mathcal{D}$ s.t. $(D, G)$ is $\varepsilon$-approximate pure equilibrium for any $\varepsilon < \frac{1}{2}q(\mathbb{E}_{\mathcal{P}_1}[D^*] - \mathbb{E}_{\mathcal{P}_2}[D^*])$.*

*Proof.* We define an alternative generator $G^*$ s.t. $\mathcal{P}_{G^*} = \mathcal{P}_1$. Given any discriminator $D$, the payoff gain of $G$ by switching strategy to $G^*$ is

$$
\begin{aligned}
&F(D, G) - F(D, G^*) \\
&= (1-q)(\mathbb{E}_{\mathcal{P}_{\text{real}}}[D] - \mathbb{E}_{\mathcal{P}_1}[D]) + q(\mathbb{E}_{\mathcal{P}_0}[D] - \mathbb{E}_{\mathcal{P}_2}[D]) - (\mathbb{E}_{\mathcal{P}_{\text{real}}}[D] - \mathbb{E}_{\mathcal{P}_2}[D]) \\
&= q(\mathbb{E}_{\mathcal{P}_1}[D] - \mathbb{E}_{\mathcal{P}_2}[D])
\end{aligned}
\tag{6}
$$

Given any discriminator $D$, the payoff gain of $D$ by switching strategy to $D^*$ is

$$
\begin{aligned}
&F(D^*, G) - F(D, G) \\
&= \mathbb{E}_{\mathcal{P}_{\text{real}}}[D^*] - (1-q)\mathbb{E}_{\mathcal{P}_1}[D^*] - q\mathbb{E}_{\mathcal{P}_2}[D^*] \\
&\quad - (\mathbb{E}_{\mathcal{P}_{\text{real}}}[D] - (1-q)\mathbb{E}_{\mathcal{P}_1}[D] - q\mathbb{E}_{\mathcal{P}_2}[D]) \\
&= \mathbb{E}_{\mathcal{P}_{\text{real}}}[D^*] - \mathbb{E}_{\mathcal{P}_1}[D^*] + q(\mathbb{E}_{\mathcal{P}_1}[D^*] - \mathbb{E}_{\mathcal{P}_2}[D^*]) \\
&\quad - (\mathbb{E}_{\mathcal{P}_{\text{real}}}[D] - \mathbb{E}_{\mathcal{P}_1}[D]) - q(\mathbb{E}_{\mathcal{P}_1}[D] - \mathbb{E}_{\mathcal{P}_2}[D]) \\
&= d_{\mathtt{W}}(\mathcal{P}_{\text{real}}, \mathcal{P}_1) + q(\mathbb{E}_{\mathcal{P}_1}[D^*] - \mathbb{E}_{\mathcal{P}_2}[D^*]) \\
&\quad - (\mathbb{E}_{\mathcal{P}_{\text{real}}}[D] - \mathbb{E}_{\mathcal{P}_1}[D]) - q(\mathbb{E}_{\mathcal{P}_1}[D] - \mathbb{E}_{\mathcal{P}_2}[D])
\end{aligned}
\tag{7}
$$

By Definition 6, $(D, G)$ cannot be $\varepsilon$-approximate equilibrium for

$$
\varepsilon < \max\left(F(D, G) - F(D, G^*), F(D^*, G) - F(D, G)\right)
\tag{8}
$$

since otherwise at least one of $D$ and $G$ will gain more than $\varepsilon$ by changing its strategy to $D^*$ or $G^*$. Therefore, we are interested in lower bounding $\min_D \max\left(F(D, G) - F(D, G^*), F(D^*, G) - F(D, G)\right)$. Note that the minimum can only be achieved when $F(D, G) - F(D, G^*) = F(D^*, G) - F(D, G)$, where we have

$$
\begin{aligned}
LHS &= q(\mathbb{E}_{\mathcal{P}_1}[D] - \mathbb{E}_{\mathcal{P}_2}[D]) \\
&= d_{\mathtt{W}}(\mathcal{P}_{\text{real}}, \mathcal{P}_1) + q(\mathbb{E}_{\mathcal{P}_1}[D^*] - \mathbb{E}_{\mathcal{P}_2}[D^*]) - (\mathbb{E}_{\mathcal{P}_{\text{real}}}[D] - \mathbb{E}_{\mathcal{P}_1}[D]) - q(\mathbb{E}_{\mathcal{P}_1}[D] - \mathbb{E}_{\mathcal{P}_2}[D]) \\
&= RHS
\end{aligned}
\tag{9}
$$

and we have

$$
\begin{aligned}
2LHS &= d_{\mathtt{W}}(\mathcal{P}_{\text{real}}, \mathcal{P}_1) + q(\mathbb{E}_{\mathcal{P}_1}[D^*] - \mathbb{E}_{\mathcal{P}_2}[D^*]) - (\mathbb{E}_{\mathcal{P}_{\text{real}}}[D] - \mathbb{E}_{\mathcal{P}_1}[D]) \\
&\geq q(\mathbb{E}_{\mathcal{P}_1}[D^*] - \mathbb{E}_{\mathcal{P}_2}[D^*])
\end{aligned}
\tag{10}
$$

where the last inequality is due to

$$
\begin{aligned}
\mathbb{E}_{\mathcal{P}_{\text{real}}}[D] - \mathbb{E}_{\mathcal{P}_1}[D] &\leq \sup_{D \in \mathcal{D}} \mathbb{E}_{\mathcal{P}_{\text{real}}}[D] - \mathbb{E}_{\mathcal{P}_1}[D] \\
&= d_{\mathtt{W}}(\mathcal{P}_{\text{real}}, \mathcal{P}_1)
\end{aligned}
\tag{11}
$$

Therefore,

$$
\min_D \max\left(F(D, G) - F(D, G^*), F(D^*, G) - F(D, G)\right)
$$

$$
\geq \frac{1}{2}q\left(\mathbb{E}_{\mathcal{P}_1}[D^*] - \mathbb{E}_{\mathcal{P}_2}[D^*]\right)
\tag{12}
$$

$\square$

**Remark.** *This result may be of independent interest.*

**Lemma 9.** *Consider the set $S_{\mathrm{OOD}} = \{x \in supp(\mathcal{P}_2) : \min_{y \in supp(\mathcal{P}_1) \cup supp(\mathcal{P}_{\mathrm{real}})} \|x - y\| \geq 1\}$. If $\mathcal{P}_2(S_{\mathrm{OOD}}) \geq 1 - q'$ for some $q' \in [0,1]$, then we have $\mathbb{E}_{\mathcal{P}_2}[D^*] \leq q'$.*

*Proof.* The value of $D$ on $\mathrm{supp}(\mathcal{P}_2) \setminus (\mathrm{supp}(\mathcal{P}_{\mathrm{real}}) \cup \mathrm{supp}(\mathcal{P}_1))$ does not affect $\mathbb{E}_{\mathcal{P}_{\mathrm{real}}}[D] - \mathbb{E}_{\mathcal{P}_1}[D]$, therefore we only need to ensure that $D^*$ satisfies the Lipschitz assumption. It is easy to see that $D^*(x)$ can be 0 for all $x \in S_{\mathrm{OOD}}$. Since $\mathcal{P}_2(S_{\mathrm{OOD}}) \geq 1 - q'$, we know that $\mathbb{E}_{\mathcal{P}_2}[D^*] \leq q'$. □

Therefore, we know that $(G, D)$ is $\varepsilon$-approximate pure equilibrium only for $\varepsilon \geq \frac{1}{2} q \left( \mathbb{E}_{\mathcal{P}_1}[D^*] - q' \right)$. Moreover, when $\mathcal{P}_{\mathrm{real}} = \mathcal{P}_1$, it is impossible to distinguish between $\mathcal{P}_{\mathrm{real}}$ and $\mathcal{P}_1$ and thus $\mathcal{D}^*$ contains function $D$ that output $D(x) = 1$ for all $x \in \mathrm{supp}(\mathcal{P}_{\mathrm{real}})$. Thus $\varepsilon \geq \frac{1}{2} q (1 - q')$.

Now we lower bound $q$, based on Assumption 1 and 2.

**Lemma 10.** $q \geq \left( \frac{1}{L\sqrt{2}} \right)^p \frac{\exp(-\frac{1}{2}B^2)}{\Gamma(p/2 + 1)}$.

*Proof.* Suppose for some $h \in \mathbb{R}^p$ we have $G(h) \in S_{\mathrm{OOD}}$, then for all $\|h' - h\|$ we have $\|G(h') - G(h)\| \leq L \|h' - h\| \leq 1$. Therefore, $G(h') \in \mathrm{supp}(\mathcal{P}_2)$. Therefore, $h'$ within the ball centered at $h$ with radius $1/L$ will all have $G(h') \in \mathrm{supp}(\mathcal{P}_2)$, and thus

$$
\begin{aligned}
q &\geq \frac{1}{(2\pi)^{p/2}} \exp(-B^2/2) \frac{\pi^{p/2}}{\Gamma(p/2 + 1)} (1/L)^p \\
&= \left( \frac{1}{L\sqrt{2}} \right)^p \frac{\exp(-\frac{1}{2}B^2)}{\Gamma(p/2 + 1)}
\end{aligned}
\tag{13}
$$

□

Plugging this lower bound back to the original bound for $\varepsilon$ leads to the final result in Theorem 7.

## C  Experimental Settings

### C.1  Choice of OOD

Clean in-distribution data may not always accessible in real-world applications, however, given the access of the target model, one may make inference about the type of data the model is trained on, e.g., whether it is a face recognition model or digit classification model, etc. This offers us the chance to make use of public available data (OOD) of the same type. FRED does not assume the overlap of the label space between OOD and the private data, as the OOD is only accessed during the GAN training stage.

Performance of FRED given different OOD is shown in Table 7, and the distance between OOD and the private distribution is calculated in Optimal Transport Dataset Distances (OTDD) (Alvarez-Melis & Fusi, 2020), which gives a meassurement of dataset distance even if the label sets are completely disjoint. As we can see, by using the in-distribution data, which in this case is CIFAR-10, FRED achieves ASR=0.01 while ACC is 0.87. Even when using OOD data which has a much larger distance, FRED can still achieve a comparable ASR and a even better ACC. The only exception is using Caltech-256 as OOD, the ASR is relatively higher compared with other settings. One possible reason is that Caltech-256 only has 30,609 samples, which is smallest dataset among the four datasets. GAN may not be well-trained on this dataset.

Given the fact that State-of-the-art MI provides could provide more advanced techniques, which relax the dependency between the target model and OOD (Struppek et al., 2022), FRED can also be benefit from it and leads to a even better performance.

|  | CIFAR-10 ⇒ CIFAR-10 | STL-10 ⇒ CIFAR-10 | Tiny-ImageNet ⇒ CIFAR-10 | Caltech-256 ⇒ CIFAR-10 |
|---|---|---|---|---|
| # Classes | 10 | 10 | 200 | 257 |
| OTDD | 324.3 | 3486.48 | 4068.28 | 3844.61 |
| ACC* | 0.96 | 0.33 | 0.74 | 0.57 |
| ACC | 0.87 | 0.85 | 0.89 | 0.80 |
| ASR | 0.01 | 0.05 | 0.02 | 0.24 |

Table 7: Results of FRED when using different OOD datasets. ACC* gives corresponding transfer accuracy respectively, and OTDD gives the distribution shift. The target model is trained on CIFAR-10 under IAB attack.

## C.2 Backdoor Attacks Implementation Details

We evaluate nine different kinds of backdoor attacks in all-to-one settings (the target model will misclassify all other classes' samples patched with the trigger as the target class), including the hidden trigger backdoor attack (Hidden) (Saha et al., 2020), input-aware backdoor (IAB) attack (Nguyen & Tran, 2020a), WaNet (Nguyen & Tran, 2020b), $L_0$ invisible ($L_0$ inv) (Li et al., 2020a), $L_2$ invisible ($L_2$ inv) (Li et al., 2020a), the frequency invisible smooth (Smooth) attack (Zeng et al., 2021), trojan watermark (Troj-WM) (Liu et al., 2018b), trojan square (Troj-SQ) (Liu et al., 2018b), and blend attack (Chen et al., 2017).

On CIFAR, we test all the nine attacks listed above. Note that initially, Hidden can only work in one-to-one attack settings where the goal is to fool one class with the trigger, thereby resulting in a low ASR in all-to-one settings. To address this issue, we manually increase the norm bound to 50/255 with one round of fine-tuning of a pre-trained clean model to achieve an acceptable ASR. However, the ASR of Hidden on GTSRB is still less than 10%, hence, we exclude it from evaluation on GTSRB. On PubFig, we adopt Trojan watermark (Troj-WM), Trojan square (Troj-SQ), and blend attack. The implementations of each attack follow the original works which propose them. The adopted trigger and the target label on each dataset is visualized in Fig. 5.

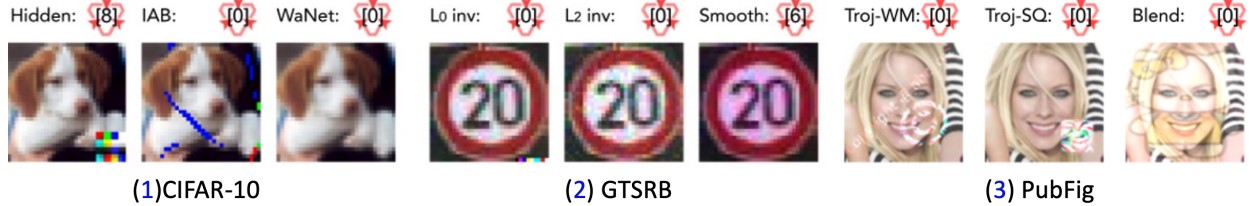

Figure 5: Datasets and examples of backdoor attacks incorporated. We consider three different datasets in this work: (1) CIFAR-10, (2) GTSRB, and (3) PubFig. Nine different backdoor attack triggers are included in the experimental part as listed. Above, we also show the target label used during the evaluated attacks (e.g., Hidden targeting at label 8 of the CIFAR-10 dataset).

## C.3 FreD Implementation Details

In our experiments, we choose the batch size $B$ in Algorithm 1 to be 40, max iterations $N = 4500$, learning rate $\alpha_2 = 1e - 3$. We choose learning rate $\alpha_1 = 2e - 3$ for CIFAR-10 and $\alpha_1 = 2e - 2$ for the rest. For parameters in Equation 4, we choose $\lambda_1 = 1000$ following the prior works on MI. We find $\lambda_3 = 1000, \lambda_4 = 10$ work well across different datasets and models. The best value for $\lambda_2$ is task-dependant, and chose using grid search that yields the smallest value of $L_{\text{prior}} + L_{\text{cl}} + L_{\text{mp}} + L_{\text{dp}}$.

In our experiments, we use 8 NVIDIA GeForce RTX 2080 Ti. The synthesize base set generation takes approximately 3 hours for private dataset CIFAR-10 and GTSRB, and 7 hours for PubFig. The computational time of the backdoor removal depends on the downstream defense method used. For I-BAU, it takes 6.82 s on average on CIFAR-10, 7.84 s on GTSRB and 13.2s on PubFig for a single iteration, and can effectively mitigate backdoor trigger within 100 iterations.

# D More FreD Boosted Defense Results

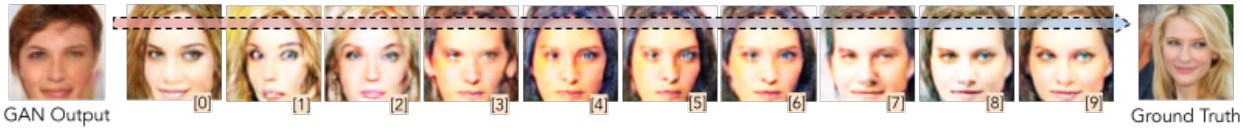

Figure 6: Examples of model-specific synthesize by FreD at first 10 iterations for Identity 12 in PubFig dataset. Rightmost in the lower row is the real image of this identity from PubFig.

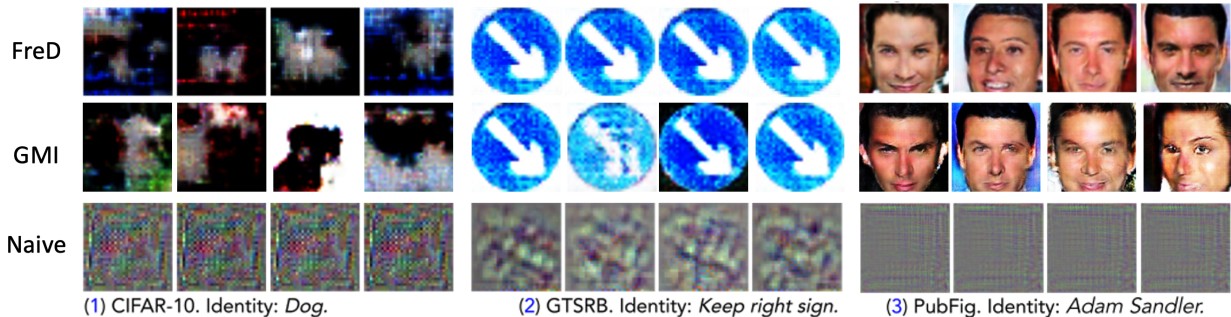

Figure 7: Examples of images obtained by FreD and naive MI. Each subplot represents randomly generated samples for the same class. The upper row shows images generated by FreD and the lower row shows images generated by naive MI.

More results of FreD boosted backdoor defense are given in Table 8,10, which FreD yields the best performance besides the clean baseline. One interesting finding is that FreD achieves a lower ASR and comparable ACC than baseline utilizing clean data when defending against the $L_2$ inv attack performed on the CIFAR-10 dataset. This may be because the model is overfitted to the clean training samples, and samples generated by FreD reduce the degree of overfitting with more abundant features.

To provide a better understanding of how FreD works, we give the images generated by FreD in the first 10 iterations for Identity 12 in PubFig dataset. As shown in Figure 6, appearance of the generated fake person gradually changed to be closer to the real identity.

We also conduct additional experiments to provide results on recent backdoor attacks, namely Blind(Bagdasaryan & Shmatikov, 2021) and SleeperAgent(Souri et al., 2022), to enhances the comprehensiveness of our evaluation. Specifically, we utilized the open-source implementation [2] of these two attacks. It's worth noting that neither of these attacks was originally evaluated on the GTSRB dataset, and their performance was found to be poor. The ASR of SleeperAgent remained consistently low (below 10%), while Blind could not effectively balance ASR and benign ACC [3], where in either case, the ASR was very low or the ACC was very low. Hence, we excluded the evaluation of these two attacks on the GTSRB dataset.

| | $L_0$ inv | | | | | | | $L_2$ inv | | | | | | | Smooth | | | | | | |
|---|---|---|---|---|---|---|---|---|---|---|---|---|---|---|---|---|---|---|---|---|---|
| | Initial | Clean | OOD | Naive | GMI | FreD | CLP | Initial | Clean | OOD | Naive | GMI | FreD | CLP | Initial | Clean | OOD | Naive | GMI | FreD | CLP |
| ACC | 0.93 | 0.89 | 0.47 | 0.44 | 0.70 | **0.76** | 0.69 | 0.94 | 0.90 | 0.45 | 0.87 | 0.89 | **0.90** | 0.67 | 0.93 | 0.83 | 0.48 | 0.54 | 0.83 | **0.83** | 0.23 |
| ASR | 0.97 | 0.15 | 0.07 | 0.09 | 0.09 | **0.06** | 0.06 | 0.99 | 0.07 | 0.12 | 0.06 | 0.06 | **0.01** | 0.05 | 0.95 | 0.18 | 0.02 | 0.20 | 0.20 | **0.18** | 0.86 |
| | Wanet | | | | | | | IAB | | | | | | | Hidden | | | | | | |
| | Initial | Clean | OOD | Naive | GMI | FreD | CLP | Initial | Clean | OOD | Naive | GMI | FreD | CLP | Initial | Clean | OOD | Naive | GMI | FreD | CLP |
| ACC | 0.94 | 0.91 | 0.84 | 0.23 | 0.79 | **0.81** | 0.75 | 0.94 | 0.82 | 0.11 | 0.34 | 0.84 | **0.85** | 0.70 | 0.76 | 0.89 | 0.11 | 0.66 | 0.86 | **0.89** | 0.35 |
| ASR | 0.99 | 0.01 | 0.32 | 0.32 | 0.03 | **0.03** | 0.05 | 0.99 | 0.03 | 0 | 0.08 | 0.06 | **0.05** | 0.03 | 0.88 | 0.09 | 0 | 0.16 | 0.13 | **0.09** | 0.94 |
| | Troj-SQ | | | | | | | Troj-WM | | | | | | | Blend | | | | | | |
| | Initial | Clean | OOD | Naive | GMI | FreD | CLP | Initial | Clean | OOD | Naive | GMI | FreD | CLP | Initial | Clean | OOD | Naive | GMI | FreD | CLP |
| ACC | 0.94 | 0.81 | 0.51 | 0.14 | 0.71 | **0.71** | 0.71 | 0.94 | 0.80 | 0.50 | 0.23 | 0.74 | **0.75** | 0.66 | 0.94 | 0.80 | 0.85 | 0.65 | 0.76 | **0.79** | 0.46 |
| ASR | 1.0 | 0.02 | 0.07 | 0.37 | 0.28 | **0.06** | 0.05 | 1.0 | 0.04 | 0.11 | 0.22 | 0.02 | **0.01** | 0.58 | 0.99 | 0.05 | 0.71 | 0.05 | 0.06 | **0.06** | 0.28 |

Table 8: Results of FreD boosted backdoor unlearning on CIFAR-10.

---

[2] https://github.com/THUYimingLi/BackdoorBox
[3] https://github.com/ebagdasa/backdoors101/issues/17)

| | SleeperAgent | | | | | | | Blind | | | | | | |
|---|---|---|---|---|---|---|---|---|---|---|---|---|---|---|
| | Initial | Clean | OOD | Naive | GMI | FreD | CLP | Initial | Clean | OOD | Naive | GMI | FreD | CLP |
| **ACC** | 0.94 | 0.91 | 0.30 | 0.81 | 0.85 | **0.86** | 0.86 | 0.67 | 0.79 | 0.16 | 0.35 | 0.44 | **0.54** | 0.50 |
| **ASR** | 0.91 | 0.02 | 0.31 | 0.10 | 0.10 | **0.02** | 0.08 | 0.78 | 0.09 | 0.23 | 0.25 | 0.10 | **0.07** | 0.20 |

Table 9: Results of FRED boosted backdoor unlearning on CIFAR-10 on recent backdoor attacks SleeperAgent(Souri et al., 2022) and Blind (Bagdasaryan & Shmatikov, 2021).

| | Troj-WM | | | | | | | Troj-SQ | | | | | | | Blend | | | | | | |
|---|---|---|---|---|---|---|---|---|---|---|---|---|---|---|---|---|---|---|---|---|---|
| | Initial | Clean | OOD | Naive | GMI | FreD | CLP | Initial | Clean | OOD | Naive | GMI | FreD | CLP | Initial | Clean | OOD | Naive | GMI | FreD | CLP |
| **ACC** | 0.92 | 0.86 | 0.06 | 0.13 | 0.83 | **0.83** | 0.01 | 0.92 | 0.84 | 0.02 | 0.18 | 0.74 | **0.78** | 0.01 | 0.91 | 0.88 | 0.06 | 0.12 | 0.83 | **0.84** | 0.03 |
| **ASR** | 1.0 | 0.03 | 0.04 | 0.23 | 0.10 | **0.03** | 0.82 | 1.0 | 0.04 | 0.002 | 0.01 | 0.06 | **0.06** | 0.89 | 1.0 | 0.44 | 0.06 | 0.78 | 0.82 | **0.52** | 0.93 |

Table 10: Results of FRED boosted backdoor unlearning on PubFig.

# E  FreD Combined with Other Backdoor Defenses

While FRED shows satisfying results when combined with I-BAU, which is one of the most popular backdoor defense framework, in this paper we also show the application of FRED combined with other backdoor defense techniques. Note that we focus on model-level cleansing in this paper. Specifically, we supply Neural Cleanse (Wang et al., 2019) and Tabor (Guo et al., 2019) with FRED synthesized data. As shown in Table 11 and 12, FRED outperforms OOD, Naive, GMI; and even achieves better results than Clean baseline under some settings.

| | IAB | | | | | | Smooth | | | | | |
|---|---|---|---|---|---|---|---|---|---|---|---|---|
| | Initial | Clean | OOD | Naive | GMI | FreD | Initial | Clean | OOD | Naive | GMI | FreD |
| **ACC** | 0.95 | 0.89 | 0.89 | 0.88 | 0.89 | 0.89 | 0.93 | 0.87 | 0.81 | 0.86 | 0.85 | 0.87 |
| **ASR** | 0.99 | 0.03 | 0.08 | 0.06 | 0.06 | 0.04 | 0.95 | 0.43 | 0.69 | 0.89 | 0.59 | 0.45 |
| | L0 inv | | | | | | L2 inv | | | | | |
| | Initial | Clean | OOD | Naive | GMI | FreD | Initial | Clean | OOD | Naive | GMI | FreD |
| **ACC** | 0.93 | 0.91 | 0.80 | 0.81 | 0.87 | 0.89 | 0.94 | 0.91 | 0.82 | 0.81 | 0.94 | 0.88 |
| **ASR** | 0.97 | 0.03 | 0.85 | 0.91 | 0.05 | 0.04 | 1.00 | 0.31 | 0.63 | 0.59 | 0.88 | 0.28 |
| | Troj-SQ | | | | | | Troj-WM | | | | | |
| | Initial | Clean | OOD | Naive | GMI | FreD | Initial | Clean | OOD | Naive | GMI | FreD |
| **ACC** | 0.94 | 0.88 | 0.69 | 0.89 | 0.72 | 0.91 | 0.94 | 0.86 | 0.77 | 0.75 | 0.85 | 0.85 |
| **ASR** | 1.00 | 0.08 | 0.98 | 0.99 | 0.10 | 0.00 | 1.00 | 0.09 | 0.98 | 0.98 | 0.16 | 0.07 |

Table 11: Results of FRED combined with Neural Cleanse on defending against various attacks on CIFAR-10.

# F  FreD against Adaptive Attacks

Another interesting case is the attacker may have knowledge about the pre-trained GAN and its corresponding training data, and design an adaptive attack to surpass our proposed defense. To investigate such setting, we designed the following attack pipeline:

1) Selecting a more promising source class: When performing a one-to-one attack, the attacker can choose a source class (i.e., the class to be poisoned) exhibiting greater similarity to the OOD data. For example, when targeting a GTSRB dataset-trained target model, the attacker may select class 6 ('80 lifted') as the source class, which has a similar pattern to class 7 in the OOD (TSRD) dataset.

2) Designing a trigger pattern closer to the OOD distribution: To achieve this, we utilize G(z) as our trigger, where $z = \arg\max_z D(G(z) + \text{source image})$, where D is the discriminator of the GAN and G is the generator. The rationale behind this approach is that we expect the poisoned source image to exhibit higher confidence under the Discriminator, making it easier to be generated by GAN.

We compare our designed adaptive attack to all the eight non-adaptive attacks we evaluate in this paper. As the baseline. As shown by Figure 8 below, the attack performance (ASR) of the adaptive procedure is lower compared to the non-adaptive ones without any defense. The reason is that this trigger is constrained to be on a specific embedding space, limiting its attack capability. However, when comparing 'After Defense' ASR of FreD, the adaptive attack indeed demonstrates better attack performance than the non-adaptive baseline, but the improvement is marginal. It is important to note that the current design is a simple and intuitive

|  | IAB | | | | | | Smooth | | | | | |
|---|---|---|---|---|---|---|---|---|---|---|---|---|
|  | Initial | Clean | OOD | Naive | GMI | FreD | Initial | Clean | OOD | Naive | GMI | FreD |
| **ACC** | 0.93 | 0.90 | 0.89 | 0.89 | 0.89 | 0.89 | 0.93 | 0.91 | 0.85 | 0.85 | 0.84 | 0.86 |
| **ASR** | 0.95 | 0.03 | 0.10 | 0.06 | 0.06 | 0.04 | 0.95 | 0.35 | 0.93 | 0.84 | 0.66 | 0.48 |
|  | **L0 inv** | | | | | | **L2 inv** | | | | | |
|  | Initial | Clean | OOD | Naive | GMI | FreD | Initial | Clean | OOD | Naive | GMI | FreD |
| **ACC** | 0.93 | 0.91 | 0.83 | 0.80 | 0.85 | 0.88 | 0.94 | 0.88 | 0.82 | 0.77 | 0.86 | 0.87 |
| **ASR** | 0.97 | 0.07 | 0.56 | 0.94 | 0.03 | 0.01 | 1.00 | 0.11 | 0.97 | 0.25 | 0.47 | 0.21 |
|  | **Troj-SQ** | | | | | | **Troj-WM** | | | | | |
|  | Initial | Clean | OOD | Naive | GMI | FreD | Initial | Clean | OOD | Naive | GMI | FreD |
| **ACC** | 0.94 | 0.89 | 0.81 | 0.86 | 0.81 | 0.90 | 0.94 | 0.88 | 0.80 | 0.81 | 0.84 | 0.88 |
| **ASR** | 1.00 | 0.13 | 0.99 | 0.77 | 0.00 | 0.00 | 1.00 | 0.40 | 0.94 | 0.95 | 0.20 | 0.16 |

Table 12: Results of FreD combined with Tabor on defending against various attacks on CIFAR-10.

example. FreD could be vulnerable to a more advanced adaptive attack, which is one of the limitations of this paper. Further improvements will be explored in future work.

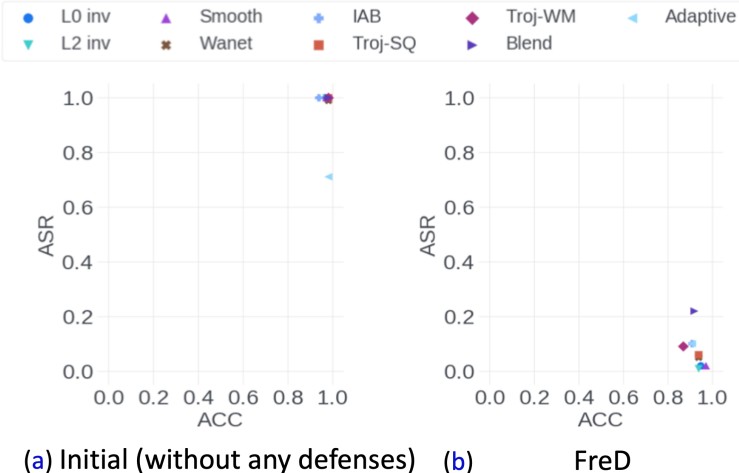

(a) Initial (without any defenses)  (b)  FreD

Figure 8: Comparison of the attack performance of non-adaptive attacks ($L_0$ *inv*, $L_2$ *inv*, IAB, Troj-WM, Troj-SQ, Wanet, Blend) and adaptive attacks (Adaptive). Figure on the left gives attack results without any defense, and figure on the right gives results of attack performance after defense with FreD.

