# OpenReview forum: "Turning a Curse into a Blessing: Enabling In-Distribution-Data-Free Backdoor Removal via Stabilized Model Inversion"
_TMLR — Accepted by TMLR_

### Review · Reviewer_BweS · 2023-05-11

**Summary Of Contributions:**

This paper combines the advances in model inversion techniques and backdoor removal techniques and show that, effective defenses can still benefit from the strong data-reliant defenses in the lack of in-distribution data. Empirical results show that the proposed attack consistently performs best across different tasks compared to the existing baselines that remove backdoors based on real or synthetic datasets.

**Audience:**

Yes

**Claims And Evidence:**

Yes

**Requested Changes:**

I do not have major concerns about this paper. A few comments that I think the authors should address is:
1. explain why the adversarial fine-tuning is interesting in practice, given the poor performance against adversarial examples compared to current state-of-the-art in RobustBench. This part will not diminish the contribution of this paper, as other baselines also perform quite poorly, instead the question is, whether this results are interesting enough to be included in the paper.
2. The authors said that minimize the euclidean distance between the synthetic examples and the poisoned samples and found that these synthetic samples are still not recognized as containing backdoors. I am wondering why Euclidean distance a useful metric here? Will cosine similarity be more help? A discuss on this will be helpful.
3. For poisoning GAN model, the authors mentioned that unless the attacker can inject enormously large fraction of samples, the GAN model is free from backdoors. But it seems even with 50% poisoning ratio, the backdoor detection rate is still as low as 2%. I am not sure why GAN can contain backdoors at large ratios such as 50% (2% detection rate is negligible number to me)? I am raising this because I am not sure if there any bugs in the implementation that caused this (seems somewhat intuitive that, when large number of samples are carefully backdoor, the GAN model should contain backdoors).


**Strengths And Weaknesses:**

Strength:
1. Although the proposed approach combines existing techniques to remove backdoors from a poisoned model, the design is not trivial with some interesting technical challenges and the paper dealt with well.
2. The empirical performance is strong and consistent, while other attacks typically have a larger variance.

Weakness:
1. The performance in adversarial fine-tuning is rather poor and it is clear whether this result can be relevant in practice. Mainly, the robustness against adversarial examples are far below the current state-of-the-art.
3. The comparison to backdoor defenses based on data sanitization defenses (not backdoor removal from a given poisoned model) is missing.

---

> ### Author Response · Authors · 2023-06-23
> **Robustness against adversarial examples are far below the current state-of-the-art.**
>
> >*Why the adversarial fine-tuning is interesting in practice, given the poor performance against adversarial examples compared to current state-of-the-art in RobustBench? (whether this results are interesting enough to be included in the paper.)*
>
> **Re:** Thanks for raising the question! We have this paragraph to show potential application of FreD, however, as FreD is not specifically designed for adversarial attacks, it is possible that its performance does not beat current state-of-the-art in RobustBench. **And we have removed this paragraph based on your suggestion**.

---

> ### Author Response · Authors · 2023-06-23
> **Why Euclidean distance a useful metric?**
>
> >*The authors said that minimize the euclidean distance between the synthetic examples and the poisoned samples and found that these synthetic samples are still not recognized as containing backdoors. I am wondering why Euclidean distance a useful metric here? Will cosine similarity be more help? A discussion on this will be helpful.*
>
>
> **Re:** The intuition here is to **encourage GAN to generate trigger pattern by maximizing** the pixel-level similarity between GAN generated images and poison images. We adopted MSE as our loss function because it’s commonly used for pixel-level recovery in the literature. While cosine similarity can also be used to measure the pixel-level similarity, it is not easy to optimize as MSE. **We have included the explanations in the revised manuscript to improve clarity.**

---

> ### Author Response · Authors · 2023-06-23
> **Why GAN can contain backdoors at large ratios such as 50% (2% detection rate is negligible number)?**
>
> >*It seems even with 50% poisoning ratio, the backdoor detection rate is still as low as 2%. I am not sure why GAN can contain backdoors at large ratios such as 50% (2% detection rate is negligible number to me)? I am raising this because I am not sure if there any bugs in the implementation that caused this (seems somewhat intuitive that, when large number of samples are carefully backdoor, the GAN model should contain backdoors).*
>
> **Re:** We apologize for the confusion in Figure 3. **Trigger detection rate refers to the percentage of images that were detected containing triggers among all generated images, while the poison rate refers to the percentage of poison images for the single target class only**. As only generated images that belong to the target class are possible to contain triggers, the overall detection rate is low.
>
> **We have changed the axis label to ‘Single-Class Poison Rate’ in the revised manuscript based on your comments.**

---

> ### Author Response · Authors · 2023-06-23
> **Comparison to data sanitization based defenses?**
>
> >*The comparison to backdoor defenses based on data sanitization defenses (not backdoor removal from a given poisoned model) is missing.*
>
> **Re:** Thanks for bringing up this interesting question! **As we discussed in the related works (Section A)**, existing backdoor defenses can be divided into two main categories: data-level and model-level. **Data sanitization based defense is an example of the data-level defenses and require access to the entire training data. We would like to clarify that we focus on model-level defenses in this paper**. In this type of defenses, the attacker aims to **remove backdoors from a given poisoned model** and  **cannot** access the entire training set.

---

### Review · Reviewer_HdcH · 2023-05-24

**Summary Of Contributions:**

The submission "Turning a Curse into a Blessing: Enabling In-Distribution-Data-Free Backdoor Removal via Stabilized Model Inversion" is concerned with backdoor attacks against image classification models. One possibility to defend against backdoor attacks is to use a backdoor-removal defense. These defenses, such as via trigger synthesis, e.g. I-BAU, often require clean data. This submission instead uses a model inversion approach, using a (clean) GAN to generate approximate data that can then be used to run back-removal via trigger synthesis. This approach is evaluated on a number of classical backdoor attacks and datasets.

**Audience:**

No

**Claims And Evidence:**

No

**Requested Changes:**

* Run the evaluation in a setting where direct comparisons are possible: Train a GAN on the same poisoned dataset and use that model for model inversion.
* Include AutoAttack with 16/255 for the adversarial finetuning. Another option would be to remove that section entirely.
* Fix Tables 10,11, Figure 1.

**Strengths And Weaknesses:**

Overall, this submission, is quite extensive in the way it approaches the topic at hand. The writing was a bit hard to understand for me, not due to grammar, but because the intent in each section was not always clear to me, and it was hard to follow along with this investigation.

I have a number of questions and comments that will list below, in no particular order:
* Fundamentally, the experimental evaluation for this work does not convince me. To circumvent the reliance on clean samples for backdoor removal, a clean dataset from a close domain is used - to train a clean GAN model instead. A larger part of the submission is then allocated to discussing why this might still be sensible, but, to me, the only relevant experiment here is the following: Use the same poisoned dataset that is used to train the model to train a GAN. Then use this GAN as a prior for the model inversion. This is also the only experiment that is directly comparable to other backdoor defenses which only have access to this dataset. This setting is covered briefly in Sec. 4.1, but why is the evaluation section not based on it? In the current submission, this feels like an add-on, instead of a core part of the argument for the sensibility of this defense.
* The paper experiments with a battery of backdoor attacks, but almost all of these are quite old, and most modern defenses perform reasonably well on these attacks. Usually, I would not feel inclined to suggest another comparison, but comparing with a battery of older does not make sense to me from a security perspective. What about newer attacks, like Blind Backdoors or Sleeper Agent?
* As an aside, it was surprising to me that Hidden Trick Backdoors, the only attack that was not included in Table 4, is also the only attack in the (broken) appendix Table 10 where the proposed approach does not win.
* Further, I don't think the experiments concerning adversarial finetuning add meaningful insights to this paper. Performance is close to random on CIFAR-10, even after this defense, and even though the strongest attack, which appears to be AutoAttack 16/255 is left out.
* Figure 1 is explained only quite late paper and quite hard to understand before then. It would be better if the caption and axes descriptions for this figure were self-contained, if it is not described in the text on related pages.


Minor:
* Typos: "Most of backdoor removal techniques", "can lead to the state-of-the-art data-free", "Note that CLP fails defensing"
* What are the compute requirements for all parts of this backdoor defense pipeline taken together?

---

> ### Author Response · Authors · 2023-06-23
> **Run the evaluation in a setting where direct comparisons are possible: Train a GAN on the same poisoned dataset and use that model for model inversion.**
>
> >*Fundamentally, the experimental evaluation for this work does not convince me. To circumvent the reliance on clean samples for backdoor removal, a clean dataset from a close domain is used - to train a clean GAN model instead. A larger part of the submission is then allocated to discussing why this might still be sensible, but, to me, the only relevant experiment here is the following: Use the same poisoned dataset that is used to train the model to train a GAN. Then use this GAN as a prior for the model inversion. This is also the only experiment that is directly comparable to other backdoor defenses which only have access to this dataset. This setting is covered briefly in Sec. 4.1, but why is the evaluation section not based on it? In the current submission, this feels like an add-on, instead of a core part of the argument for the sensibility of this defense.*
>
> **Re:** We believe this might be a misunderstanding. We would like to emphasize that **most of the existing backdoor removal strategies require extra clean validation sets (referred to as base sets in this paper) from the same training distribution** (refer to Section A). In contrast, as discussed in the main paper, our focus is to enable backdoor removal in scenarios where clean in-distribution data is not available. This unique characteristic of our approach allows for a plug-and-play defense and also benefits from advancements in  defense techniques that rely on base sets.
>
> Furthermore, **our evaluation setting reflects real-world scenarios, where the training data of pre-trained GANs often differs from the target model's training set in terms of distribution** (e.g., pre-trained GAN is usually built with large, open data, yet the target model could be built by some stakeholder with proprietary datasets). We also considered out-of-distribution data with various distributional shifts from the target-model training data to enable a comprehensive evaluation (refer to Section C.1).
>
> **Our baselines provide comparison from different perspectives**: 1) Clean is the original backdoor removal method with **an extra clean in-distribution** set provided, serving as the upper bound for defense performance. 2) OOD, Naive, GMI share the same assumption as our defense that the defender can leverage clean publicly available data from a related domain. This setting is practical in real-world scenarios considering the large amount of public data available online. 3) CLP is the state-of-the-art backdoor removal defense that is not base-set-reliant compared to this type of baselines, our method offers a unique plug-and-play benefit.

---

> ### Author Response · Authors · 2023-06-23
> **Include AutoAttack with 16/255 for the adversarial finetuning. Another option would be to remove that section entirely.**
>
> >*Further, I don't think the experiments concerning adversarial finetuning add meaningful insights to this paper. Performance is close to random on CIFAR-10, even after this defense, and even though the strongest attack, which appears to be AutoAttack 16/255 is left out.*
>
> **Re:** Thanks for raising the question! We have this paragraph to show potential application of FreD, however, as FreD is not specifically designed for adversarial attacks, it is possible that its performance does not beat current state-of-the-art in RobustBench. **And we have removed this paragraph based on your suggestion.**

---

> ### Author Response · Authors · 2023-06-23
> **(minor) Fix Tables 10,11, Figure 1. Fix typos. Add computation requirements for the entire pipeline.**
>
> >*Figure 1 is explained in later paper, hard to understand before then. Make the caption and axes description self-contained.
> Typos: "Most of backdoor removal techniques", "can lead to the state-of-the-art data-free", "Note that CLP fails defensing".
> What are the compute requirements for all parts of this backdoor defense pipeline taken together?*
>
> **Re:** Thanks for your suggestion! **We’ve updated our manuscript accordingly in the revised pdf (Figure 1 in Section 1, typos, computation requirement in Section C.4).**

---

> ### Author Response · Authors · 2023-06-23
> **Backdoor attacks evaluated are old. What about newer ones?**
>
> >*The paper experiments with a battery of backdoor attacks, but almost all of these are quite old, and most modern defenses perform reasonably well on these attacks. Usually, I would not feel inclined to suggest another comparison, but comparing with a battery of older does not make sense to me from a security perspective. What about newer attacks, like Blind Backdoors or Sleeper Agent?*
>
> **Re:** Thank you for your suggestion! **We have included the evaluation of Blind Backdoor and SleeperAgent on the CIFAR-10 dataset in the revised manuscript (Section D).** For these evaluations, we utilized the open-source implementation (https://github.com/THUYimingLi/BackdoorBox) of these two attacks. It's worth noting that neither of these attacks were originally evaluated on the GTSRB dataset, and their performance was found to be poor. The ASR of SleeperAgent remained consistently low (below 10%), while Blind could not effectively balance ASR and benign ACC (as discussed in this thread https://github.com/ebagdasa/backdoors101/issues/17), where in either case, the ASR was very low or the ACC was very low. Hence, we excluded the evaluation on the GTSRB dataset.

---

> ### Author Response · Authors · 2023-06-23
> **Hidden is not included in Table4, and FreD does not win on Hidden in Table10.**
>
> >*As an aside, it was surprising to me that Hidden Trick Backdoors, the only attack that was not included in Table 4, is also the only attack in the (broken) appendix Table 10 where the proposed approach does not win.*
>
> **Re:** Thanks for pointing this out, and **we have fixed the broken table issue in the revised pdf. We would also like to clarify that FreD does achieve the best performance on Hidden on the CIFAR-10 dataset** (Table 9 in revised pdf, Table 10 in original one). While its performance matches that of the clean baseline, it is important to note that the clean baseline relies on a stronger assumption of having a clean in-distribution set available, making it an upper bound for performance comparison.
>
>
> For evaluation on GTSRB dataset (Table 3), **we excluded Hidden because the ASR is less than 10% as explained in Appendix Section C.2**: Initially, Hidden can only work in one-to-one attack settings where the goal is to fool one class with the trigger, thereby resulting in a low ASR in all-to-one settings. To address this issue, we manually increase the norm bound to $50/255$ with one round of fine-tuning of a pre-trained clean model to achieve an acceptable ASR. However, the ASR of Hidden on GTSRB is still less than 10%, hence, we exclude it from evaluation on GTSRB.

---

### Review · Reviewer_xTon · 2023-06-10

**Summary Of Contributions:**

The paper proposes a technique to remove backdoors from a classifier by using model inversion attacks to generate new un-backdoored data. While prior work has used model inversion attacks for this purpose in the past, this work improves upon that work by taking advantage of developments in the model inversion field, using a model inversion attack that uses on a GAN pretrained on OOD data.

**Audience:**

Yes

**Claims And Evidence:**

No

**Requested Changes:**

Discussion of the potential for adaptive attacks

Improved clarity of the proof of theorem 7

Discussion that GAN backdoors might be improvable

Adding discussion on these points would lead me to raise the claims and evidence score and recommend accept

**Strengths And Weaknesses:**

Strengths:

The paper is generally written and well cited. There are places that are rather confusing, like the structure of Section 3, the placement of the GAN poisoning experiments of Section 4, and the theoretical results in the appendix, but otherwise, I enjoyed reading this paper.

The idea of the paper is intuitive and combines multiple fields of adversarial machine learning. FreD gets strong results.

Weaknesses:

The reliance on a pretrained, unpoisoned GAN is rather strong. I think their OOD baseline is an appropriate one that addresses this assumption, but there may be other ways to take advantage of this GAN to improve clean performance.

Unless something really clever is happening in training, robustness to a 50% poisoning rate for any model is implausible, as in Figure 3. I think it’s fine that the paper has these results, but they seem to be more a result of the limited work on backdooring GANs in the literature, rather than a fundamental robustness of GAN training. I’d recommend the authors present these results with a bit more caveats attached. In particular, it’s unlikely that the FreD technique would be robust to an adaptive poisoning attack on the GAN.

Comments:
Theorem 3 and Theorem 7 appear to be the informal/formal version of the same theorem, but the paper never says this. The proof also needs more intuitive guidance.

I wonder if an adversary with access to the pretrained GAN could design a more effective backdoor attack on the paper’s proposed approach. For example, the attack could design a backdoor attack which operates on some features that are not included in the in-distribution data, but are included in the OOD distribution. This would allow the GAN to have backdoored outputs.

Overall assessment:
Interestingness: Given that this work builds off of model inversion attacks in the literature, and there is also a literature on backdoor sanitization, I think these subcommunities would find this paper interesting. There are a couple limitations that impact this interestingness, including the reliance on the GAN and the prior work which has proposed a similar idea.

Correctness: The results of training the GAN on poisoned data do not seem to be the results of a worst case poisoning attack. The proof of Theorem 3 is also not written clearly enough for me to understand the intuition.  I am also worried about robustness to adaptive attacks. Small changes to improve the clarity of Theorem 3 and attach more caveats to the results of Figure 3, as well as some discussion of adaptive attacks, would address my concerns here.

---

> ### Author Response · Authors · 2023-06-23
> **Discussion of the potential for adaptive attacks.**
>
> >*Reliance on an unpoisoned GAN is strong. …  I wonder if an adversary with access to the pretrained GAN could design a more effective backdoor attack on the paper’s proposed approach. For example, the attack could design a backdoor attack which operates on some features that are not included in the in-distribution data, but are included in the OOD distribution. This would allow the GAN to have backdoored outputs.*
>
> **Re:** Thanks for bringing up this interesting case. To investigate the adaptive attack setting wherein the attacker has knowledge about the pre-trained GAN and its corresponding training data, **we designed the following attack pipeline**:
>
> **1) Selecting a more promising source class:** When performing a one-to-one attack, the attacker can choose a source class (i.e., the class to be poisoned) exhibiting greater similarity to the OOD data. For example, when targeting a GTSRB dataset-trained target model, the attacker may select class 6 ('80_lifted') as the source class, which has a similar pattern to class 7 in the OOD (TSRD) dataset.
>
> **2) Designing a trigger pattern closer to the OOD distribution:** To achieve this, we utilize G(z) as our trigger, where z = \argmax_z D(G(z) + source_image), where D is the discriminator of the GAN and G is the generator The rationale behind this approach is that we expect the poisoned source image to exhibit higher confidence under the Discriminator, making it easier to be generate by GAN.
>
> We consider the (non-adaptive)  L0_inv attack as the baseline, which represents the best-performing attack prior to the defense. As shown by Table 3 in the revised paper, the attack performance (ASR) of the adaptive procedure is lower compared to the non-adaptive one without any defense. The reason is that this trigger is constrained to be on a specific embedding space, limiting its attack capability. However, when comparing ‘After Defense’ ASR of FreD, the adaptive attack indeed demonstrates better attack performance than the non-adaptive baseline, but the impovement is marginal It is important to note that the current design is simple and intuitive example. FreD could be vulnerable to a more advanced adaptive attack. **We have acknowledged this limitation in the revised paper, and further improvements will be explored in future work.**

---

> ### Author Response · Authors · 2023-06-23
> **Discussion that GAN backdoors might be improvable.**
>
> >*Unless something really clever is happening in training, robustness to a 50% poisoning rate for any model is implausible, as in Figure 3. I think it’s fine that the paper has these results, but they seem to be more a result of the limited work on backdooring GANs in the literature, rather than a fundamental robustness of GAN training. I’d recommend the authors present these results with a bit more caveats attached. In particular, it’s unlikely that the FreD technique would be robust to an adaptive poisoning attack on the GAN.*
>
> **Re:** We apologize for the confusion in Figure 3. **Trigger detection rate refers to the percentage of images that were detected containing triggers among all generated images, while the poison rate refers to the percentage of poison images with the target-class images only**. As only generated images that belong to the target class are possible to contain triggers, the overall detection rate is low.
>
> **We have changed the axis label to ‘Single-Class Poison Rate’ in the revised manuscript based on your comments.**

---

> ### Author Response · Authors · 2023-06-23
> **Improved clarity of the proof of theorem 7.**
>
> >*Theorem 3 and Theorem 7 appear to be the informal/formal version of the same theorem. Intuition of Theorem 3? Clarity of Theorem 7? Requested changes: Improved clarity of the proof of theorem 7.*
>
> **Re:** Thanks for the suggestion! **We have modified our manuscript in Section B based on your comments. We added the general idea, interpretation, intuition and proof sketch for clarity in the revised manuscript.**
>
> ---
> **General Idea**: game-theoretic interpretation of GAN. From a game theory perspective, the generator and discriminator in GAN are engaged in a two-player zero-sum game. In game theory, a Nash Equilibrium is a state in which neither player can improve their own outcome by changing their strategy, assuming the other player keeps their strategy fixed. In the context of GANs, a Nash Equilibrium is achieved when the Generator produces data that is indistinguishable from real data, and the Discriminator is no better than random guessing at telling the difference between real and generated data. In Theorem 3 (or Theorem 7), we show that when the generator’s learned distribution contains a significant probability mass on backdoored images, which is out-of-distribution (OOD) data, the equilibrium cannot be established between the generator and discriminator, which forces the generator to reduce the probability mass on the backdoored images.
>
> **Interpretation of Theorem 7.** To interpret the statement of Theorem 7, one can regard $P_1$ as a clean distribution (not necessarily $P_{\text{real}}$), and $P_2$ as a distribution that contains backdoor data on its support. Since backdoored images are separated from clean images (i.e., out-of-distribution (OOD) data), we can assume that all backdoored images are within the set $S_{\mathrm{OOD}} = \{x \in \text{supp}(P_2): \min_{y \in \text{supp}(P_1) \cup \text{supp}(P_{\text{real}})} \lVert x-y \rVert \ge 1\}$. The above theorem thus states that no equilibrium could be achieved if $P_G$ has non-negligible density on $S_{\mathrm{OOD}}$.
>
> **Intuition**: The theorem essentially reveals the tension between a generator's goal to trick the discriminator and its propensity to generate backdoored (OOD) data. In simple terms, think of $P_1$ as a distribution representing normal, clean images, and $P_2$ representing malicious, backdoored images. The generator mixes these distributions to create $P_G$, which it uses to try to fool the discriminator. Now, if the $P_G$ mixture has a significant probability of sampling backdoored (OOD) images from $P_2$, it is much easier for the discriminator to spot the differences between the generated and real data. Hence, there exists a better move for the generator to increase its "payoff" by sampling less from $P_2$, hence an equilibrium becomes unattainable under such $P_G$ (captured by the upper bound of $\epsilon$). In essence, the generator is in a quandary: including backdoored data in the mixture makes it easier for the discriminator to win, which is contrary to what the generator wants.
>
> **Proof sketch for Section B.1**: In Lemma 8, we derive the upper bound of the degree of equilibrium $\epsilon$ when $P_G = (1-q) p_1 + q p_2$ in terms of the best discriminator in distinguishing between $P{\text{real}}$ and $P_1$, i.e., $D^* = \arg\max_{D \in \mathcal{D}} \mathbb{E}{P_\text{real}} [D] - \mathbb{E}_{P_1}[D]$. Then in Lemma 9 and 10, we further bound different terms in the upper bound of $\epsilon$ derived in Lemma 8 under Assumption 1 and 2 stated in the main text.
> (We are sorry that some of the latex code is not displayed properly, please refer to our revised manuscript.)

---

> ### Author Response · Authors · 2023-06-23
> **The prior work has proposed a similar idea?**
>
> **Re:** As mentioned in our abstract and introduction, previous work (DeepInspect) did a simple combination of an existing MI technique and backdoor removal, suffering limited performance. **This paper is the first to study how to actively design new MI techniques that benefit backdoor removal.**

---

### Review · Reviewer_cAzf · 2023-07-21

**Summary Of Contributions:**

The paper proposes a new idea of using removing backdoors from a poisoned model. The idea is to use a generative model trained on near-distribution data and then use an existing backdoor removal technique to reduce the effect of the backdoors. The authors show that by modifying the sampling procedure they can improve the quality of the model and also reduce the effect of backdooring.


**Audience:**

Yes

**Claims And Evidence:**

No

**Requested Changes:**

Improved the adaptive attack.
Improve the structure and flow of the text in the paper

**Strengths And Weaknesses:**


Strength:
- The paper uses new sampling techniques to improve the existing backdoor removal approaches.
- The experimental results in the evaluated setting are strong

Weaknesses:
- The paper can be a bit hard to understand, especially in Section 3. I suggest the authors start by formulating their approach and then explaining it. In the current structure the authors keep mentioning the changes they will do to Equation 3 in the text. It might make it easier to follow the text, if instead they write down the final equation and then explain each component.

- Table 2 can be improved by making it a grid so it would be easier to understand how the parameters affect each other. At the moment the numbers in two columns are significantly different and it is not clear to me how they affect each other.

- While the work relaxes the requirements for in-distribution clean data, now it requires a generative model trained on non-backdoored data. This is a significant assumption and it is not very well discussed in the work.

- One of the main limitations of this approach is that it is heavily reliant on the generative model. As the authors show in Theorem 3 and 7, if the distribution of the training data used to train the generative model and the target model is different, then the generative model won’t generate samples from that distribution. This can be a very limiting factor. The authors add a few experiments showcasing when the distribution of the generative model is different from the target model, however, the experiments are fairly limited. It would be interesting to add some additional experiment showing the effect of different datasets (e.g, using MedMNIST datasets)

- Also the author considered an adaptive adversary which is very good and appreciated, however, the analysis is very limited and shallow. The authors should provide reasoning why such attacks are best attacks that an adaptive adversary can perform. For example for the first attack the authors can show the effect of selecting different classes on the attack success rate. The second attack is not clear from the text. What exactly does `z = arg max  (D(z) + source image)` do? Discriminator does not get z as input also how does output of the discriminator is summed with an image ?  Also the authors argue that they use L_0 inv because it is the best attack, but I am not sure how they are claiming that ? Based on Table 5 it seems this is not true. In any case it would be much better to show the results for all of the attacks since actually this is the main interesting setting.

- The authors should expand more on an adaptive adversary, for example assuming the adversary has access to the generative model, then it can try to force the backdoors to be very similar to the distribution of the generative model. While I think this is what the second attack is supposed to be doing, I couldn’t understand their approach.


- The number is Tables 12 and 13 are in percentages while the rest of the paper is not, it is better to make the paper consistent.
Overall it seems the additional text added to the paper based on the reviewer is not well integrated to the text and it makes it very difficult to understand the work.

---

> ### Author Response · Authors · 2023-07-25
> **Improve the structure and flow of the text in the paper**
>
> > The paper can be a bit hard to understand, especially in Section 3. I suggest the authors start by formulating their approach and then explaining it. In the current structure the authors keep mentioning the changes they will do to Equation 3 in the text. It might make it easier to follow the text, if instead they write down the final equation and then explain each component.
>
> **Re:** Our original structure begins with studying three factors that may affect backdoor removal performance (Section 3.1), where each finding motivates a specific loss term design we proposed in this paper. We summarized these loss terms together and proposed our method FreD in Section 3.2, while the final optimization is given in equation 4.
>
> **In response to your suggestion, we have included the final equation at the beginning of Section 3 for improved readability. For further details, please refer to the updated manuscript.**

---

> ### Author Response · Authors · 2023-07-25
> **Improve the adaptive attack.**
>
> > Also the author considered an adaptive adversary which is very good and appreciated, however, the analysis is very limited and shallow. The authors should provide reasoning why such attacks are best attacks that an adaptive adversary can perform.
>
> **Re:** Thank you for your feedback. We would like to clarify that our paper primarily focuses on defense , and as such, we have thoroughly evaluated it against various existing attacks. The design presented here serves as an illustrative example to showcase the potential of adaptive attacks, as stated in our paper: **"It is important to note that the current design is a simple and intuitive example" (last paragraph of Section 4.1).** It might not the best attack that an adaptive adversary can perform. We acknowledge that there could be room for further improvements in adaptive attacks, and we believe that exploring potential enhancements would be valuable for future research and worth to be a separate paper.
>
> > The second attack is not clear from the text. What exactly does z = arg max (D(z) + source image) do? Discriminator does not get z as input also how does output of the discriminator is summed with an image ? …The authors should expand more on an adaptive adversary, for example assuming the adversary has access to the generative model, then it can try to force the backdoors to be very similar to the distribution of the generative model. While I think this is what the second attack is supposed to be doing, I couldn’t understand their approach.
>
> **Re:**
> We believe that there is a misunderstanding here. We only presented one potential adaptive attack. The first and second steps combined together is our designed adaptive attack. We apologize that there was a typo in the equation of the second step, which might have caused the misunderstanding. The correct equation should be z = \argmax_z D(G(z) + source_image), where D is the discriminator of the GAN and G is the generator The rationale behind this approach is that we expect the poisoned source image to exhibit higher confidence under the Discriminator, making it easier to be generate by GAN. **We have updated the equation in the revised manuscript.**
>
> > Also the authors argue that they use L_0 inv because it is the best attack, but I am not sure how they are claiming that ? Based on Table 5 it seems this is not true. In any case it would be much better to show the results for all of the attacks since actually this is the main interesting setting.
>
> **Re:**  **We want to clarify that the adaptive attack proposed here is a seperate attack where the trigger is get from optimizing the objective z = \argmax_z D(G(z) + source_image), it is not adapted from L_0 inv attack.**  Hence, in Table 3, the non adaptive results can be any one of the attack results from Table 5 (e.g., L_2 inv, Smooth, Wanet, etc). We picked L_0 inv for presentation because it achieves the highest ASR (100%) when there is no defense. We realized this could be misleading. Hence, to improve the clarity of the result presentation, we have included a plot (Figure 3) in the revised manuscript to visualize the before- and after-defense attack performance for **all** the non-adaptive attacks we evaluated against in this paper and the new designed adaptive attacks.

---

> ### Author Response · Authors · 2023-07-25
> **Assumption Discussion**
>
> > While the work relaxes the requirements for in-distribution clean data, now it requires a generative model trained on non-backdoored data. This is a significant assumption and it is not very well discussed in the work.
>
> **Re:This assumption is discussed in Section C.1**, “Clean in-distribution data may not always accessible in real-world applications, however, given the access of the target model, one may make inference about the type of data the model is trained on, e.g., whether it is a face recognition model or digit classification model, etc. This offers us the chance to make use of public available data (OOD) of the same type. \AlgName does not assume the overlap of the label space between OOD and the private data, as the OOD is only accessed during the GAN training stage.” Besides, **according to our empirical study in Section 4.1,** trigger detection rate remains low (0.02) even when poison rate is increased to an uncommonly large ratio (50% of the images from the target classes are poisoned). Based on these findings, **we do not have a hard restriction on non-backdoored GAN training data: even if backdoors exist in GAN's training data, it would be hard for them to take effect unless using a substantial poison rate, which becomes unrealistic.** Also, recent backdoor data detection methods [1] have achieved great efficacy. They can serve as pre-processing steps to screen out poisoned samples in GAN’s training set.
>
> [1] Ma, Wanlun, et al. "The" Beatrix''Resurrections: Robust Backdoor Detection via Gram Matrices." arXiv preprint arXiv:2209.11715 (2022).

---

> ### Author Response · Authors · 2023-07-25
> **Add additional experiment showing the effect of different GAN datasets**
>
> > One of the main limitations of this approach is that it is heavily reliant on the generative model. As the authors show in Theorem 3 and 7, if the distribution of the training data used to train the generative model and the target model is different, then the generative model won’t generate samples from that distribution. This can be a very limiting factor. The authors add a few experiments showcasing when the distribution of the generative model is different from the target model, however, the experiments are fairly limited. It would be interesting to add some additional experiment showing the effect of different datasets (e.g, using MedMNIST datasets)
>
> **Re:** Thank you for the comment. We would like to kindly remind the reviewers that Section C.1 presented a comprehensive study of how distributional shifts between GAN’s training data and target model’s training data would impact performance (Table 8), where we have examined GAN trained on different datasets including CIFAR-10, STL-10, Tiny-ImageNet, Caltech-256.
>
> As we discussed in Section C.1, “Clean in-distribution data may not always accessible in real-world applications, however, given the access of the target model, one may make inference about the type of data the model is trained on, e.g., whether it is a face recognition model or digit classification model, etc. This offers us the chance to make use of public available data (OOD) of the same type.” As the OOD is picked by the attacker to maximize the chance of succesful attack, we argue that it does not make sense to assume the attacker would choose the GAN’s training data to be a entirerly different type of data from the target model’s training data. While we appreciate the suggestion of the MedMNIST dataset, it is worth noting that it is of different type from the CIFAR data and supports a different prediction task.

---

> ### Author Response · Authors · 2023-07-25
> **Improve Table 12 and 13.**
>
> > The number is Tables 12 and 13 are in percentages while the rest of the paper is not, it is better to make the paper consistent. Overall it seems the additional text added to the paper based on the reviewer is not well integrated to the text and it makes it very difficult to understand the work.
>
> **Re: We have updated the number of Table 12 and 13 to be not in percentage in the revised manuscript.**

---

### Author Response · Authors · 2023-06-23
**Summary**

Dear Reviewers,

We sincerely appreciate all Your insightful comments and questions. We are delighted that our work was recognized as interesting (xTon, BweS) and extensive (HdcH), the proposed method is non-trivial (BweS) and has shown strong results (xTon, BweS). We are also grateful for acknowledging the quality of our writing and references (xTon).

In response to your valuable feedback, we have made the following improvements to our work and uploaded a revised manuscript:
***
- We have included a detailed discussion on the potential for adaptive attacks.
- We have conducted additional experiments to provide results on recent backdoor attacks, namely Blind and Sleeper Agent attacks, as suggested. This addition enhances the comprehensiveness of our evaluation.
- After careful consideration, we have removed the section on adversarial fine-tuning.
- We have provided a thorough explanation of our empirical study on MI ]not recovering backdoors, including clarifications on Figure 3 and the intuition behind using Euclidean distance.
- We have added general ideas, interpretation, intuition and a proof sketch for Theorem 7.
- We have emphasized that our evaluation setting, where the GAN is trained on out-of-distribution (OOD) data, is practical and reflects real-world scenarios. It is common for the training data of pre-trained GANs to differ from the target model's training set in terms of distribution.
- We have addressed minor issues such as broken tables, typos, and improved figure captions.
***

Please notify us if You have any further suggestions, we would be glad to discuss them.


With Gratitude,
Authors of Paper991

---

### Decision · Action_Editors · 2023-08-03

**Recommendation:** Accept with minor revision

**Comment:**

The reviewers found the experimental results strong, but not entirely convincing. The reason here is that while the proposed method seems to work well against known attacks, there seems no fundamental reason to believe that the method would prevent more carefully designed attacks.

The reviewer's opinion is split on this paper, and a majority of the reviewers would have enjoyed to see a deeper investigation of adaptive attacks. However, the reviewers agree that the claims wrt. to the investigated attacks are clear and justified by evidence.


For the camera-ready version, please:
- avoid overclaims on 'no clean data needed',
- remove the red annotations
- fix formatting issues (e.g. there are whitespaces after equation (2) and (3), etc.)
- the new figures (e.g. the grid) could perhaps be formatted a bit more nicely, and rearranged (currently page 9 is half-empty), so that all fits again in 12 pages

**Audience:**

Yes, this could be interesting for parts of the TMLR audience (although some reviewers noted that experts in the backdoor community would have enjoyed if more stronger attacks would have been explored more deeply in the paper).

**Claims And Evidence:**

The paper proposes a method to remove backdoors from a poisoned model. The method requires a generative model trained on (clean) near-distribution data and then use an existing backdoor removal technique to reduce the effect of the backdoors.

The reviewers found that the paper gives evidence that the proposed defense does defend against the baseline attacks that are investigated in the paper.

It is clearly stated in the submission that the method requires access to a generator trained on clean data. While this might be strong assumption, it is plausible in certain applications. However, the submission sometimes overclaims the premises of the method (e.g. in abstract: "without requiring access to clean in-distribution data") and the authors are strongly encouraged to carefully rephrase such statements that could give a wrong impression.